# From Pixels to Tokens: A Systematic Study of Latent Action Supervision for Vision-Language-Action Models

Yihan Lin [1 2]  Haoyang Li [1 2]  Yang Li [1 2]  Haitao Shen [1 2]  Yihan Zhao [1 2]  Chao Shao [1 2]  Jing Zhang [1 3]

## Abstract

Latent actions serve as an intermediate representation that enables consistent modeling of vision-language-action (VLA) models across heterogeneous datasets. However, approaches to supervising VLAs with latent actions are fragmented and lack a systematic comparison. This work structures the study of latent action supervision from two perspectives: (i) regularizing the trajectory via image-based latent actions, and (ii) unifying the target space with action-based latent actions. Under a unified VLA baseline, we instantiate and compare four representative integration strategies. Our results reveal a formulation-task correspondence: image-based latent actions benefit long-horizon reasoning and scene-level generalization, whereas action-based latent actions excel at complex motor coordination. Furthermore, we find that directly supervising the VLM with discrete latent action tokens yields the most effective performance. Finally, our experiments offer initial insights into the benefits of latent action supervision in mixed-data, suggesting a promising direction for VLA training. Code is available at GitHub.

## 1. Introduction

Vision-Language-Action (VLA) models have emerged as a paradigm for learning generalist robotic policies (Kim et al., 2024; Black et al., 2024; Brohan et al., 2022; Zitkovich et al., 2023; Pertsch et al., 2025). By training on large-scale datasets, these methods achieve robust performance across diverse tasks and environments (Intelligence et al., 2025; Bjorck et al., 2025). However, the VLA training data are heterogeneous, spanning inconsistent action spaces from various robotic platforms and human videos (Kareer et al., 2025; Chi et al., 2024; Grauman et al., 2022). This heterogeneity may lead to performance degradation due to mismatched action semantics across domains, motivating the use of suitable intermediate representations.

Latent actions abstract motion from either visual transitions or raw actions, offering a unified supervision signal for training VLA models on heterogeneous data (Schmidt & Jiang, 2023; Lee et al., 2024). Recent VLA models have started to leverage latent actions to improve generalization (Bi et al., 2025; Bjorck et al., 2025), but their integration strategies into VLA models remain fragmented. Existing approaches range from discrete token supervision (Ye et al., 2024; Bu et al., 2025a) to auxiliary learning objectives (Chen et al., 2025b), all targeting the VLM backbone; this diversity suggests different assumptions about the role of latent actions, leaving the optimal formulation unclear. In this work, rather than evaluating latent action models themselves, we study how different latent action supervision choices affect VLA policy learning under a unified baseline.

We systematize integration strategies from two complementary perspectives, defined by the mapping direction between vision-language (VL) space and action spaces: (1) *Regularizing the Trajectory*. This forward mapping uses *image-based latent actions* as high-level visual plans to guide the VLM's decision-making. (2) *Unifying the Target Space*. This reverse mapping unifies actions into *action-based latent actions* aligned with the VLM token space, bridging semantic mismatches across heterogeneous action spaces.

Motivated by our two perspectives, we propose four distinct supervision strategies (as shown in Fig. 1). For trajectory regularization, we investigate *Implicit Representation Alignment*, *Explicit Direct Decoding*, and *Explicit Conditional Decoding*; for target unification, we consider *Action-to-Token Mapping*. To evaluate these strategies under fair conditions, we implement them within a unified VLA baseline, specifically designed with a shared backbone and action head. We evaluate these strategies across benchmarks (LIBERO and RobotWin 2.0) and real-world experiments. We identify three empirical findings: (i) image-based latent actions are more effective for long-horizon reasoning and

---

[1]School of Information, Renmin University of China, Beijing, China [2]Key Laboratory of Data Engineering and Knowledge Engineering, Beijing, China [3]Engineering Research Center of Database and Business Intelligence, Beijing, China. Correspondence to: Jing Zhang <zhang-jing@ruc.edu.cn>.

*Proceedings of the 43rd International Conference on Machine Learning*, Seoul, South Korea. PMLR 306, 2026. Copyright 2026 by the author(s).

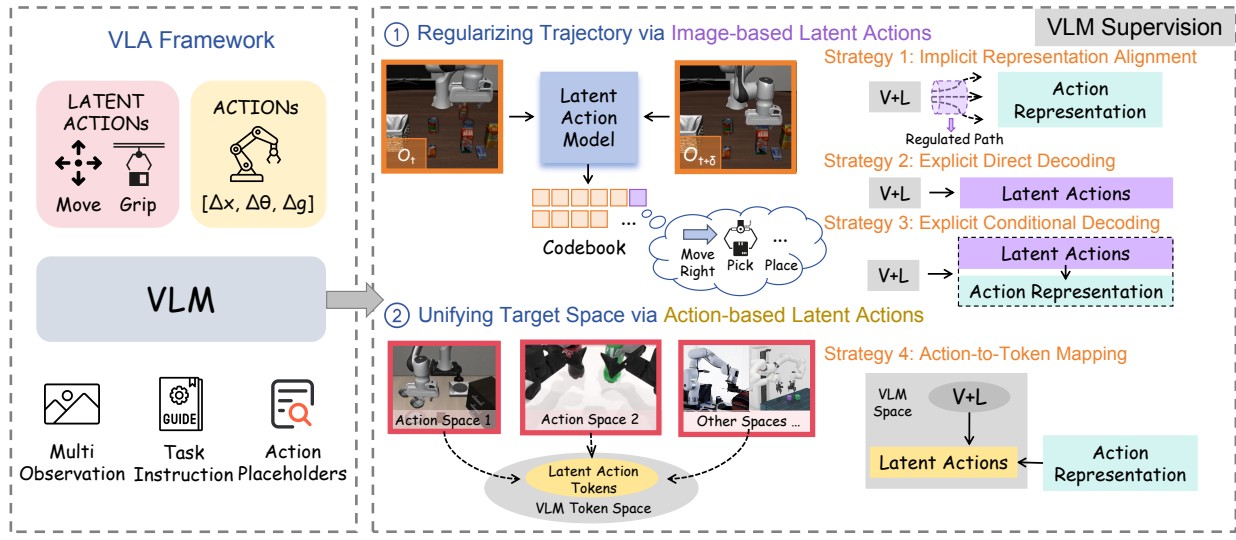

*Figure 1.* Overview of latent actions in VLA. Left: a unified VLA pipeline; right: our two perspectives and four integration strategies.

scene-level generalization, while action-based latent actions excel at motorically complex tasks; (ii) directly supervising VLMs to predict discrete latent action tokens yields the most effective performance; and (iii) latent action supervision consistently improves performance under joint training on mixed-task datasets. These results provide guidance for selecting latent action supervision strategies in VLA policy learning. Overall, the core contributions include:

- We systematically study latent action supervision under a unified VLA baseline, isolating the effects of latent action formulation and architectural integration.

- We categorize latent action supervision into two complementary perspectives and propose four integration strategies for VLA training.

- We provide extensive empirical analysis that reveals (i) formulation-task correspondence, (ii) the comparative effectiveness of different strategies, and (iii) robustness under joint training on mixed-task datasets.

## 2. Related Work

### 2.1. Latent Action Representations

Latent actions are compact representations used to structure action modeling, capturing either visual dynamics or control abstractions. Prior work broadly falls into two directions: image-based and action-based. On one hand, image-based latent actions learn a latent action model from visual transitions to capture structured changes between images, as explored in Genie (Bruce et al., 2024) and LAPO (Bu et al., 2025b). Subsequent work further studies how more effectively capture task-relevant changes in visual transitions (Nikulin et al., 2025; Zhang et al., 2025a). On the

other hand, action-based latent actions are learned from continuous action trajectories to induce compact and semantically structured action spaces. Some approaches focus on extracting recurring behavioral patterns from action sequences (Mete et al., 2024; Lee et al., 2024; Wang et al., 2024). Others discretize actions and align the resulting representations to external semantic spaces, such as MLLM token spaces (Szot et al., 2024) or downstream VLA task spaces (Zhang et al., 2025b).

### 2.2. VLA Models Integrating with Latent Actions

Recent VLA models have increasingly incorporated latent actions to structure policy learning, exploring a variety of integration strategies. Methods using image-based latent actions guide low-level control through visual abstractions (Ye et al., 2024). Some train the backbone to predict latent actions that are decoded into actions by a separate decoder (e.g., UniVLA (Bu et al., 2025a), IGOR (Chen et al., 2024)), while others jointly generate latent actions and actions within the backbone (e.g., Moto-GPT (Chen et al., 2025c), villa-x (Chen et al., 2025b)). Conversely, methods using action-based latent actions compress continuous actions into a unified latent space to bridge heterogeneous action spaces (Zhang et al., 2025b). However, prior methods entangle latent action formulation and integration with VLA architectural design. This work isolates their effects to study how latent action supervision influences policy learning.

## 3. Preliminaries

### 3.1. Vision-Language-Action Models

We consider VLA models formulated as conditional sequence generation. Given visual observations $o_t \in \mathcal{O}$

and a language instruction $\ell \in \mathcal{L}$ at timestep $t$, the objective is to learn a policy $\pi$ that predicts an action chunk $\mathrm{a}_t = \langle a_t, a_{t+1}, \ldots, a_{t+H-1} \rangle \in \mathcal{A}^{H \times m}$, where $H$ is the chunk size, $m$ is action dimension. The model is trained on a set of demonstration trajectories $\mathcal{D} = \{(o_t^i, \ell, \mathrm{a}_t^i)\}_{i=1,t=1}^{N,T_i}$, where the model minimizes the reconstruction error between the predicted chunk and the ground truth action sequence.

### 3.2. Latent Action Models

We categorize latent action models into two types based on their input modalities and modeling objectives. All latent action models are kept frozen during VLA training.

#### 3.2.1. IMAGE-BASED LATENT ACTION MODELS.

Image-based latent action models learn compact action-relevant dynamics from visual transitions. Given a temporal horizon $\delta$, a start observation $o_t$, and an end observation $o_{t+\delta}$, the model encodes the visual transition into a discrete latent action $z_t^{\text{img}}$.

$$\begin{cases} c_t^{\text{img}} = E_\theta\big([o_t; o_{t+\delta}]\big), & c_t^{\text{img}} \in \mathbb{R}^d, \\ z_t^{\text{img}} = \text{VQ}(c_t^{\text{img}}), & z_t^{\text{img}} \in \{1, \ldots, K_{img}\}^P, \\ \hat{o}_{t+\delta} = D_\theta\big([o_t; e(z_t^{\text{img}})]\big). \end{cases} \tag{1}$$

Here, $E_\theta$ and $D_\theta$ denote the encoder and decoder, $\text{VQ}(\cdot)$ is a vector-quantization operator with a learned codebook $e(\cdot)$, and $c_t^{\text{img}}$ denotes the continuous embedding, where $d$ is the embedding dimension and each timestep corresponds to $P$ discrete tokens. We pre-compute $z_t^{\text{img}}$ and $c_t^{\text{img}}$ using a latent action model fine-tuned based on UniVLA (Bu et al., 2025a) with the standard VQ-VAE objective (Van Den Oord et al., 2017), including reconstruction, codebook, and commitment terms (Appendix A.1).

#### 3.2.2. ACTION-BASED LATENT ACTION MODELS.

Action-based latent action models map continuous action chunks into a shared discrete token space for VLM supervision across datasets. Given an action chunk $\mathrm{a}_t$, the model encodes it into latent embeddings and quantizes them into discrete codes:

$$\begin{cases} c_t^{\text{act}} = E_\phi(\mathrm{a}_t), & c_t^{\text{act}} \in \mathbb{R}^{H \times d}, \\ z_t^{\text{act}} = \text{VQ}(c_t^{\text{act}}), & z_t^{\text{act}} \in \{1, \ldots, K_{act}\}^H, \\ \hat{\mathrm{a}}_t = D_\phi\big(e(z_t^{\text{act}})\big), & \hat{\mathrm{a}}_t \in \mathbb{R}^{H \times m}. \end{cases} \tag{2}$$

Here, $E_\phi, D_\phi, \text{VQ}(\cdot), d$ and $e(\cdot)$ follow the same definitions as Eq. (1). Analogously, $c_t^{\text{act}}$ and $z_t^{\text{act}}$ denote the continuous and discrete chunk-level representations, respectively. The model is trained with an action chunk reconstruction objective (with VQ commitment/codebook terms).

Building on this formulation, we propose an action-based latent action model (Fig. 2) that discretizes each action chunk

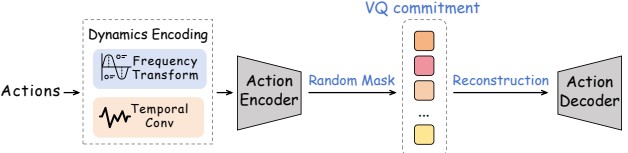

*Figure 2.* Architecture of our action-based latent action model.

into a sequence of latent tokens. This model is designed for controlled comparison rather than for optimizing standalone state-of-the-art performance. The encoder captures both coarse and fine action dynamics within a chunk by combining frequency and time-domain representations: we apply FFT to model slow, low-frequency trends over the horizon, and use 1D temporal convolutions to capture fast variations across neighboring timesteps. A Transformer encoder then integrates information across all timesteps within the chunk. In addition to reconstruction and VQ losses, we introduce a latent consistency loss $\mathcal{L}_{\text{mask}}$ that randomly masks a subset of latent timesteps, decodes from the masked latent sequence, and enforces the re-encoded latents to match the originals at masked positions. This serves as a regularization mechanism that stabilizes tokenization and improves the effectiveness of discrete action supervision. Full architectural, loss, and training details are provided in Appendix A.2.

### 3.3. Unified VLA Baseline

We build our baseline (Figure 3(a)) on Qwen3-VL-2B. Given $o_t, \ell$, VLM processes these inputs together with $H$ action placeholders. The function $f_{\text{head}}$ predicts continuous actions from the image token representations, placeholder hidden states and robot proprioceptive state $s_t$. Across all strategies, the backbone, placeholder design, aggregation function, and action head (including its inputs) are kept identical; the only difference lies in the supervision imposed on the VLM representations.

$$\begin{aligned} h_t^{\text{act}}/h_t^{\text{latent}} &= \text{Agg}\big(\{\nu_t^{(k)}\}_{k \in \mathcal{K}_{\text{layers}}}\big), \\ \hat{a}_{t:t+H-1} &= f_{\text{head}}\big([h_t^{\text{img}}; h_t^{\text{act}}; s_t]\big). \end{aligned} \tag{3}$$

Here, $h_t^{img}$ denotes the aggregated image token representations from the VLM backbone. $\nu_t^{(k)} \in \mathbb{R}^{H \times d_{hidden}}$ denotes the placeholder representation at layer $k$, where $d_{hidden}$ denotes the VLM hidden dimension. We extract placeholders from the latter half of the VLM layers, $\mathcal{K}_{\text{layers}} = \lceil N_{\text{layer}}/2 \rceil, \ldots, N_{\text{layer}}$, and stack them along the layer dimension via $\text{Agg}(\cdot)$ to obtain $h_t^{\text{act}}$. Under latent-action supervision, we denote the resulting representation as $h_t^{\text{latent}}$. Additional training details are provided in Appendix B.1.

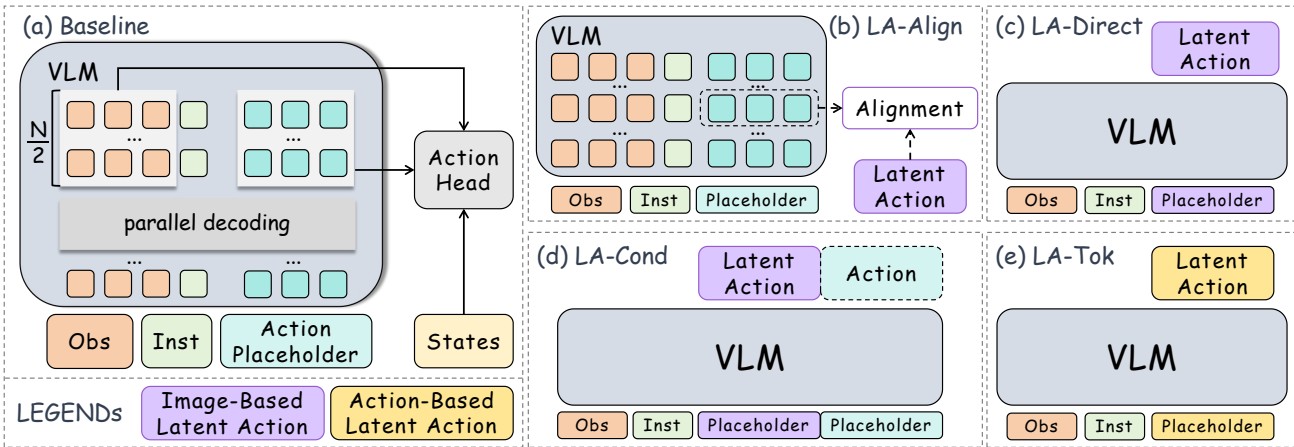

*Figure 3.* Architectural instantiations of the Baseline and four VLM supervision strategies. (a) Baseline. (b) Strategy 1 (Implicit Representation Alignment): LA-Align. (c) Strategy 2 (Explicit Direct Decoding): LA-Direct. (d) Strategy 3 (Explicit Conditional Decoding): LA-Cond. (e) Strategy 4 (Action-to-Token Mapping): LA-Tok.

## 4. Methodology

We systematically investigate how to best utilize latent actions by organizing our study around the mapping direction between continuous action spaces and the VLM token space (Fig. 1). We identify two fundamental perspectives: (1) **Regularizing the Trajectory**, where we leverage *image-based latent actions* to provide high-level visual plans that guide the VLM's reasoning process; and (2) **Unifying the Target Space**, where we employ *action-based latent actions* to inversely map continuous actions into the VLM's semantic token space. These perspectives form four distinct supervision strategies for the VLM. Fig. 3 illustrates the corresponding architectural instantiations. In panels (b-e), we only visualize the VLM-related components; all other modules and connections are shared with panel (a).

### 4.1. Formulation: Mapping Perspectives and Strategies

**Perspective 1: Image-based Latent Actions Regularize the Trajectory.** We introduce an intermediate visual plan as high-level supervision. Specifically, we use image-based latent actions to guide VLM training and investigate three integration strategies:

- **Strategy 1: Implicit Representation Alignment.** An auxiliary objective aligns internal VLM representations with latent action embeddings.

- **Strategy 2: Explicit Direct Decoding.** The VLM is directly supervised to predict latent actions as an intermediate planning representation.

- **Strategy 3: Explicit Conditional Decoding.** The VLM jointly predicts latent plans and action representations, with the latter conditioned on the former.

**Perspective 2: Action-based Latent Action Unifies the Target Space.** This perspective bridges the modality gap by reformulating the supervision target. We leverage an action-based latent action (Sec. 3.2.2) to discretize continuous actions into action-based latent actions, providing a unified target space for VLM training.

- **Strategy 4: Action-to-Token Mapping.** Discretize actions into tokens and directly supervise the VLM to predict them as the training target.

**Unified Objective.** We formulate a generalized training objective. Given the continuous action head loss $\mathcal{L}_{\text{action}}$, we introduce a latent supervision loss $\mathcal{L}_{\text{latent}}$ to regularize the VLM backbone:

$$\mathcal{L}_{\text{action}} = \mathbb{E}_t \left[ \sum_{\tau=0}^{H-1} \left\| \hat{a}_{t+\tau} - a^*_{t+\tau} \right\|_2^2 \right],$$

$$\mathcal{L} = \mathcal{L}_{\text{action}} + \lambda \mathcal{L}_{\text{latent}}. \tag{4}$$

where $\lambda$ balances the strength of the latent guidance. We detail $\mathcal{L}_{\text{latent}}$ in Sections 4.2 and 4.3. All four strategies run in a single forward pass at test time, and latent action supervision introduces no additional inference stages; detailed strategy-specific configurations are provided in Appendix B.2.

### 4.2. Image-based Latent Actions: Trajectory Regularization

In this section, we detail Strategies 1–3, which incorporate image-based latent actions as intermediate plans for trajectory regularization (Fig. 3(b–d)).

#### 4.2.1. STRATEGY 1: IMPLICIT REPRESENTATION ALIGNMENT (FIG. 3(B))

Implicit representation alignment encourages the VLM to internalize latent actions as an intermediate planning representation, inspired by prior work on aligning 3D spatial signals with VLM image tokens (Li et al., 2025). This strategy injects planning supervision by regularizing the internal features toward the ground-truth latent embedding $c_t^{\text{img}}$. Concretely, we impose an alignment constraint at a layer $k$ of the VLM (we choose 17 out of 29). We introduce a linear projection function $\phi_{align}(\cdot)$ that maps the hidden state of the action placeholder at this layer, denoted as $\nu_t^{(k)}$, into the latent action embedding space. The alignment objective (latent supervision) is defined as:

$$\mathcal{L}_{\text{latent}} = -\mathbb{E}_t\Big[\mathcal{S}\big(\phi_{align}(\nu_t^{(k)}),\, c_t^{\text{img}}\big)\Big], \qquad (5)$$

where we use cosine similarity as $\mathcal{S}(\cdot, \cdot)$.

#### 4.2.2. STRATEGY 2: EXPLICIT DIRECT DECODING (FIG. 3(C))

Explicit direct decoding supervises the VLM to predict image-based latent actions directly as discrete tokens, abstracting a supervision pattern used in prior models (e.g., UniVLA (Bu et al., 2025a), LAPA (Ye et al., 2024)). We represent the latent actions as a token sequence $z_{t:t+H-1}^{img} \in \{1, \ldots, K_{img}\}^{H \times P}$ and train the VLM to output the corresponding latent action tokens at dedicated placeholder positions. We denote the planner distribution predicted by the VLM as $\pi_{\text{vlm}}$. The probability distribution over the codebook indices is defined as:

$$\pi_{vlm}(z_{t:t+H-1}^{\text{img}} \mid o_t, \ell) = \text{Softmax}\big(\phi_{explicit}(\nu_t^{(N_{\text{layer}})})\big), \qquad (6)$$

where $\nu_t^{(N_{\text{layer}})}$ corresponds to the final-layer states of the latent action placeholders, and $\phi_{explicit}$ is a linear projection head.

The VLM is trained with explicit latent action tokens by maximizing the log-likelihood of the ground-truth sequence:

$$\mathcal{L}_{\text{latent}} = -\sum_{h=0}^{H-1} \log \pi_{vlm}(z_{t+h}^{\text{img}*} \mid o_t, \ell). \qquad (7)$$

To compute the action loss in Eq. (4), we decode the latent action representation into continuous actions. Given the latent action representation $h_t^{\text{latent}}$, the action head infers the corresponding continuous action chunk by incorporating the image token representations and proprioceptive state $s_t$:

$$\hat{a}_{t:t+H-1} = f_{\text{head}}\big([h_t^{\text{img}}; h_t^{\text{latent}}; s_t]\big). \qquad (8)$$

Overall, this design trains the VLM backbone to directly predict discrete latent action tokens, shaping its internal representations prior to continuous action decoding.

#### 4.2.3. STRATEGY 3: EXPLICIT CONDITIONAL DECODING (FIG. 3(D))

Explicit conditional decoding jointly models latent actions and action representations, where action decoding is explicitly conditioned on the predicted latent actions. This formulation abstracts a prevalent joint-generation design, in which latent actions and actions are generated with conditioning (e.g., Moto-GPT (Chen et al., 2025c), villa-x (Chen et al., 2025b)). Concretely, Strategy 3 splits the placeholders into two segments: (i) a latent segment that predicts discrete latent actions, and (ii) an action segment used to form the action representation. We implement this conditioning by placing latent placeholders before action placeholders and enforcing the dependency through the causal attention mask.

The latent segment follows the same direct decoding formulation as Eq. 6, producing $z_{t:t+H-1}^{\text{img}}$ and trained with the latent supervision objective in Eq. 7. The action placeholder states are preserved as action representations and aggregated as in Eq. 3 to form $h_t^{\text{act}}$. The action head then predicts actions by combining $h_t^{\text{img}}$, $h_t^{\text{latent}}$, $h_t^{\text{act}}$, and $s_t$:

$$\hat{a}_{t:t+H-1} = f_{\text{head}}\big([h_t^{\text{img}}; h_t^{\text{latent}}; h_t^{\text{act}}; s_t]\big), \qquad (9)$$

**Discussion.** Both Strategy 2 and Strategy 3 explicitly regularize VLM via discrete latent action supervision. Compared to Strategy 2, Strategy 3 conditions action prediction on the decoded latent plan, encouraging the VLM to first form a high-level plan and then generate low-level actions accordingly.

### 4.3. Action-based Latent Actions: Target Unification

Action-based latent actions unify the supervision target by discretizing continuous actions into a shared token space. In contrast to image-based latent actions that act as intermediate guidance, this formulation turns actions themselves into discrete prediction targets that the VLM can directly model and supervise.

#### 4.3.1. STRATEGY 4: ACTION-TO-TOKEN MAPPING (FIG. 3(E))

Strategy 4 unifies the target space by mapping a continuous action chunk $a_t$ into a discrete token sequence $z_t^{\text{act}} \in \{1, \ldots, K_{act}\}^H$ using the latent action model introduced in Section 3.2.2. The VLM is then trained to predict these discrete action tokens:

$$\pi_{\text{vlm}}(z_t^{\text{act}} \mid o_t, \ell) = \text{Softmax}\big(\phi_{tok}(\nu_t^{(N_{\text{layer}})})\big), \qquad (10)$$

where $\phi_{tok}$ is a linear projection head, $\nu_t^{(N_{\text{layer}})}$ follow the same definitions as Eq. 6. The loss of latent supervision follows the same objective as in Strategy 2 (Eq. 7). The action head deterministically maps the action representation to the continuous action chunk, i.e., $\hat{a}_t = f_{\text{head}}([h_t^{\text{img}}; h_t^{\text{latent}}; s_t])$.

*Table 1.* Success Rates (%) on the **LIBERO benchmark**. **Bold** indicates the best performance and underline indicates the second best. Green/Red numbers in parentheses show the improvement/degradation over our baseline. Results for prior methods are taken from their original papers and are provided for reference only; implementation and citation details are provided in Appendix C.1.

| METHOD | SPATIAL | OBJECT | GOAL | LONG | AVG. |
|---|---|---|---|---|---|
| OPENVLA-OFT(KIM ET AL., 2025) | **97.6** | 98.4 | **97.9** | 94.5 | **97.1** |
| $\pi_0$(BLACK ET AL., 2024) | 96.8 | 98.8 | 95.8 | 85.2 | 94.2 |
| LAPA(YE ET AL., 2024) | 73.8 | 74.6 | 58.8 | 55.4 | 65.7 |
| UNIVLA(BU ET AL., 2025A) | 96.5 | 96.8 | 95.6 | 92.0 | 95.2 |
| THINKACT(HUANG ET AL., 2025) | 88.3 | 91.4 | 87.1 | 70.9 | 84.4 |
| GR00T(BJORCK ET AL., 2025) | 94.4 | 97.6 | 93.0 | 90.6 | 93.9 |
| $\pi_0$-FAST(PERTSCH ET AL., 2025) | 96.4 | 96.8 | 88.6 | 60.2 | 85.5 |
| BASELINE | 96.6 | 97.2 | 92.8 | 85.8 | 93.1 |
| LA-ALIGN | 97.4 (↑ **0.8**%) | 98.6 (↑ 1.4%) | 97.2 (↑ **4.4**%) | 94.8 (↑ 9.0%) | 97.0 (↑ 3.9%) |
| LA-DIRECT | 97.2 (↑ 0.6%) | 98.6 (↑ 1.4%) | 95.8 (↑ 3.0%) | **96.6** (↑ **10.8**%) | **97.1** (↑ **4.0**%) |
| LA-COND | 97.0 (↑ 0.4%) | 99.4 (↑ 2.2%) | 95.8 (↑ 3.0%) | 94.2 (↑ 8.4%) | 96.6 (↑ 3.5%) |
| LA-TOK | 97.0 (↑ 0.4%) | **100.0** (↑ 2.8%) | 92.2 (↓ 0.6%) | 92.6 (↑ 6.8%) | 95.5 (↑ 2.4%) |

*Table 2.* Success Rates (%) on the **RoboTwin 2.0** benchmark. **Bold** indicates the best performance and underline indicates the second best. Green/Red numbers in parentheses show the improvement/degradation over the baseline. Results for prior methods are taken from RoboTwin2.0 leaderboard and are provided for reference only; implementation and citation details are provided in Appendix C.2.

| METHOD | MOVE PLAYINGCARD AWAY | PLACE CONTAINER PLATE | MOVE CAN POT | PICK DUAL BOTTLES | AVG |
|---|---|---|---|---|---|
| RDT(LIU ET AL., 2024) | 43 | 78 | 25 | 42 | 47.0 |
| $\pi_0$(BLACK ET AL., 2024) | 53 | 88 | 58 | 57 | 64.0 |
| ACT(ZHAO ET AL., 2023) | 36 | 72 | 22 | 31 | 40.3 |
| DP3(ZE ET AL., 2024) | 68 | 86 | **70** | 60 | 71.0 |
| BASELINE | 73 | 86 | 46 | 37 | 60.5 |
| LA-ALIGN | 78 (↑ 5%) | 88 (↑ 2%) | 55 (↑ 9%) | 61 (↑ 24%) | 70.5 (↑ 10%) |
| LA-DIRECT | 76 (↑ 3%) | **96** (↑ **10**%) | 64 (↑ 18%) | 51 (↑ 14%) | 71.8 (↑ 11.3%) |
| LA-COND | 76 (↑ 3%) | 89 (↑ 3%) | 52 (↑ 6%) | **78** (↑ **41**%) | 73.8 (↑ 13.3%) |
| LA-TOK | **89** (↑ **16**%) | 89 (↑ 3%) | **70** (↑ **24**%) | 64 (↑ 27%) | **78.0** (↑ **17.5**%) |

## 5. Experiments

Our experiments investigate the impact of latent actions on VLA policies by addressing the following three questions:

**(Q1) Formulation:** Which latent action formulation performs better: image-based latent actions or action-based latent actions?

**(Q2) Integration:** Which integration architecture is most effective for latent action supervision, i.e., which strategy yields the best performance?

**(Q3) Representation:** Does discrete token supervision provide a more effective supervision signal than continuous representation?

### 5.1. Experimental Setup

**Tasks and Benchmarks.** We evaluate on two widely used simulated benchmarks, LIBERO (Liu et al., 2023) and RoboTwin 2.0 (Chen et al., 2025a), as well as a real-world JAKA robotic arm. We follow the official evaluation proto-

cols for both simulated benchmarks. For LIBERO, we use the LIBERO-Long suite to emphasize long-horizon multi-stage manipulation. For RoboTwin 2.0, we evaluate on a 4-task subset covering both single-arm and coordinated dual-arm tasks, which represents a motorically complex setting due to higher-dimensional control and the presence of coordinated dual-arm manipulation tasks. For real-world experiments, we evaluate on JAKA across two groups of tasks: manipulation tasks comprising bowl stacking and stain wiping (Stack 2 Bowls and Wipe Stains as easier tasks, Stack 3–4 Bowls as a long-horizon setting); and pick-and-place tasks with irregular or contact-rich objects on a cluttered tabletop, evaluated under both in-domain (ID) and out-of-domain (OOD) scene configurations. We report completion scores in [0, 100] for manipulation tasks and pick-and-place success scores over 10 rollouts; additional details are provided in Appendix F.

**Implementation.** Following the strategies delineated in Sec. 4, we implement four variants: **LA-Align (S1)**, **LA-Direct (S2)**, **LA-Cond (S3)**, and **LA-Tok (S4)**. Detailed

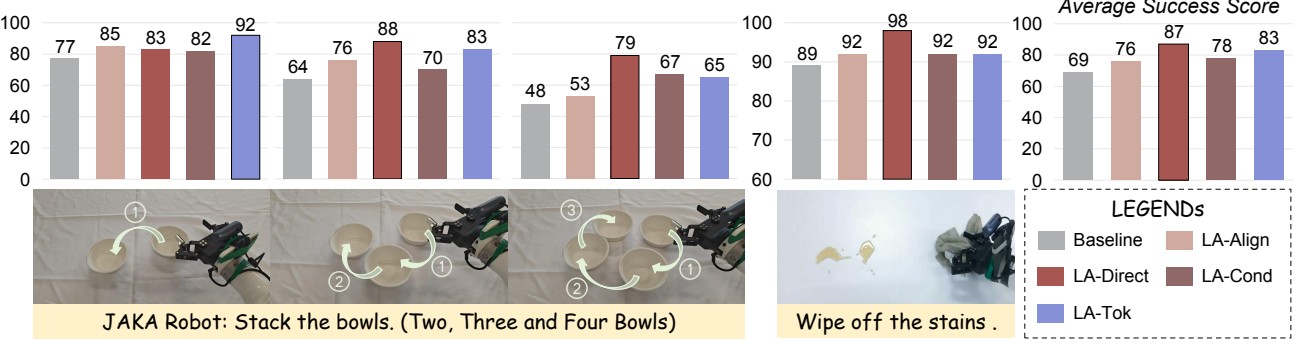

*Figure 4.* **Real-world manipulation task results.** Scores are reported on a [0,100] completion-percentage scale, details provided in Appendix F.2. Mean scores for each of the four real-world tasks across different models (10 rollouts per model-task), detailed results are in Appendix F.3.

*Table 3.* **Real-world pick-and-place task results.** Scores are reported on a [0,100] scale, details provided in Appendix F.2 and Appendix F.3. **Bold** indicates the best performance and underline indicates the second best. Green/Red numbers in parentheses show the improvement/degradation over the baseline in the average columns.

| METHOD | MANGO | | SPONGE | | BOTTLE | | AVERAGE | | |
|---|---|---|---|---|---|---|---|---|---|
| | ID | OOD | ID | OOD | ID | OOD | ID | OOD | TOTAL |
| BASELINE | **100** | 30 | 60 | 20 | **70** | 0 | 77 | 17 | 47 |
| LA-ALIGN | 90 | 80 | **90** | 60 | 60 | 30 | 80 (↑ 3) | 57 (↑ 40) | 68 (↑ 21) |
| LA-DIRECT | **100** | 70 | **90** | **80** | 60 | **50** | **83** (↑ 6) | 67 (↑ 50) | **75** (↑ 28) |
| LA-COND | 90 | **90** | **90** | **80** | 50 | 40 | 77 | **70** (↑ 53) | 73 (↑ 26) |
| LA-TOK | **100** | 60 | **90** | 70 | 60 | 20 | **83** (↑ 6) | 50 (↑ 33) | 67 (↑ 20) |

training configurations are provided in Appendix D.

## 5.2. Formulation Analysis: Image-Based vs. Action-Based Latent Actions

We do not compare the latent action models themselves, but instead study the effect of latent action supervision under a unified VLA baseline. Across all benchmarks, latent action supervision yields consistent improvements over the baseline, and even competitive with state-of-the-art VLAs on simulation benchmarks. Moreover, the two formulations exhibit different empirical advantages.

**Image-based Latent Actions for Long-Horizon Tasks.** On LIBERO-Long (Tab. 1), image-based strategies yield larger improvements (+8.4% to +10.8%) than the action-based strategy LA-Tok (+6.8%), suggesting that supervision over visual state transitions is particularly helpful for long-horizon planning. This observation is consistent with our real-world manipulation tasks (Fig. 4): on Stack 4 Bowls task, LA-Direct achieves a score of 79 compared to 48 for the baseline. These results suggest that image-based latent actions provide supervision on visual state transitions, which may encourage more coherent long-horizon behavior.

**Action-based Latent Actions for Motorically Complex Tasks.** On RoboTwin 2.0, LA-Tok achieves the best performance, improving the average success rate by +17.5% over

the baseline (Tab. 2). On real-world easier tasks (stacking 2–3 bowls and pick-and-place ID tasks, Fig. 4 and Tab. 3), it also ranks among the top methods (1st or 2nd). We hypothesize that discretizing continuous actions into tokens provides a more structured supervision signal, which may facilitate learning complex motor behaviors.

## 5.3. Integration Analysis: Optimal Strategy Selection

We analyze how different *architectural integration strategies* affect performance across tasks.

**(1) Within explicit image-based designs, Explicit Direct Decoding generally outperforms Explicit Conditional Decoding in most experiments.** LA-Direct consistently performs better on long-horizon settings such as LIBERO-Long, achieving 96.6% success compared to 94.2% for LA-Cond. On real-world tasks, LA-Direct outperforms LA-Cond by +9 points on manipulation tasks (87 vs. 78, Fig. 4) and achieves better overall performance on pick-and-place tasks (75 vs. 73, Tab. 3). These results suggest that decoupling latent plan prediction from action representation learning (LA-Direct) leads to more effective performance, while jointly modeling both representation (LA-Cond) may be more beneficial in complex settings such as RoboTwin 2.0 (Tab. 2).

**(2) Explicit supervision architectures outperform im-**

*Table 4.* Ablation on the representation of latent action supervision (continuous embedding vs. discrete token) for LA-Direct and LA-Tok.

| METHOD | SPA. | OBJ. | GOAL | LONG | AVG |
|---|---|---|---|---|---|
| BASELINE | 96.6 | 97.2 | 92.8 | 85.8 | 93.1 |
| LA-DIRECT(C) | 95.4 | 97.0 | 90 | 95.2 | 94.4 |
| LA-DIRECT | **97.2** | **98.6** | **95.8** | **96.6** | **97.1** |

| METHOD | SPA. | OBJ. | GOAL | LONG | AVG |
|---|---|---|---|---|---|
| BASELINE | 96.6 | 97.2 | **92.8** | 85.8 | 93.1 |
| LA-TOK(C) | 95.6 | 98.4 | 87.6 | 91.6 | 93.3 |
| LA-TOK | **97.0** | **100.0** | 92.2 | **92.6** | **95.5** |

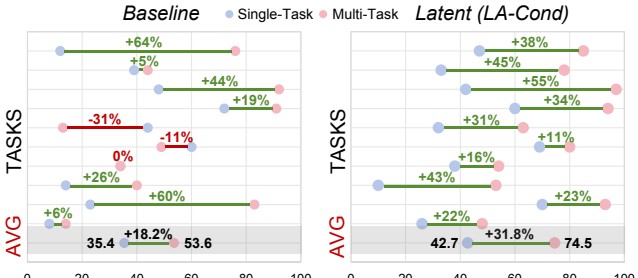

*Figure 5.* Comparing Baseline (left) and LA-Cond (right) performance on 10 RoboTwin tasks. Green/red denotes performance gain/drop under joint training relative to the baseline.

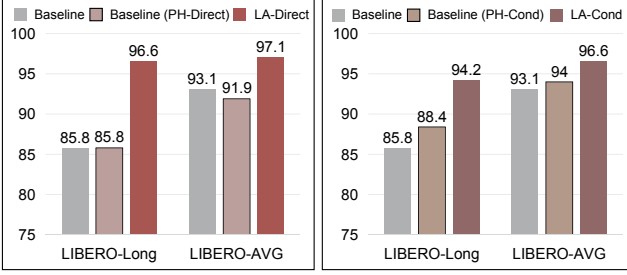

*Figure 6.* **Ablation on placeholder length (PH-L).** Extending the action placeholder yields little benefit compared to latent action integration strategies (details in Appendix E.2).

**plicit alignment.** Comparing explicit planning supervision (LA-Direct) with implicit feature regularization (LA-Align), LA-Direct consistently delivers stronger gains. On LIBERO-Long, LA-Direct reaches 96.6%, outperforming LA-Align at 94.8%. The gap is also evident in real-world manipulation tasks (Avg. 87 vs. 76, Fig. 4) and pick-and-place tasks (Avg. 75 vs. 68, Tab. 3), while results on RoboTwin show a smaller but still favorable margin. These results suggest that directly supervising the VLM output space with discrete targets provides a stronger architectural constraint than aligning representations implicitly.

**(3) Directly supervising the VLM to predict latent actions yields the most effective performance among the evaluated strategies.** Across experiments, the strongest strategies are those that directly supervise the VLM to predict latent actions (LA-Direct and LA-Tok), with LA-Direct leading on LIBERO and LA-Tok leading on RoboTwin 2.0. In real-world experiments, LA-Direct ranks first on both manipulation tasks (Avg. 87, Fig. 4) and pick-and-place tasks (Avg. 75, Tab. 3), while LA-Tok remains competitive as the second-best on manipulation tasks (83). Overall, directly supervising the VLM to predict latent actions provides a simple and consistent supervision target for VLM training. Meanwhile, the choice between image-based and action-based formulations further affects which task regimes benefit the most (Sec. 5.2).

### 5.4. Representation Analysis: Discrete Tokens vs. Continuous Representation

Motivated by the findings in Sec. 5.3(3), we focus on strategies where the VLM is directly supervised to predict latent actions, and examine whether discrete token supervision

is more effective than its continuous counterpart. To construct the continuous variants, we regress the corresponding continuous latent representations ($c^{img}$ or $c^{act}$) from the final-layer placeholder states $\nu_t^{(N_{layer})}$ using a two-layer MLP trained with an MSE loss. As shown in Tab. 4, discrete supervision consistently outperforms continuous regression, achieving +2.7% and +2.2% higher average success rates, respectively. Notably, on LIBERO-Long, the continuous variants still substantially outperform the baseline, further indicating that latent action supervision remains informative even with continuous regression.

### 5.5. Real-World Scene Generalization

Tab. 3 reports the results on real-world pick-and-place tasks under ID and OOD scene configurations. Latent action supervision consistently improves robustness over the baseline under OOD scene variations, where distractor objects and target layouts differ from the training demonstrations. Image-based strategies demonstrate a clear advantage under OOD settings, with LA-Cond achieving the highest OOD average score (70), while LA-Direct obtains the best total average score (75). The baseline degrades significantly under distribution shift, with an OOD average score of 17.

### 5.6. Evaluation of Scaling, Efficiency, and Ablation Study

**Latent actions reduce negative transfer in multi-task joint training.** Scaling VLA policies to diverse task distributions often incurs negative transfer. As illustrated in Fig. 5, the Baseline exhibits instability during joint training on 10 RoboTwin tasks: while the average performance improves, specific tasks suffer significant degradation. In contrast,

*Table 5.* Ablation on different align layer in LA-Align on LIBERO benchmarks.

| METHOD | SPA. | OBJ. | GOAL | LONG | AVG |
|---|---|---|---|---|---|
| BASELINE | 96.6 | 97.2 | 92.8 | 85.8 | 93.1 |
| LA-ALIGN(EARLY) | 94.0 | 96.2 | 91.0 | 85.2 | 91.6 |
| LA-ALIGN(FINAL) | 96.8 | **99.2** | 92.2 | 91.8 | 95.0 |
| LA-ALIGN | **97.4** | 98.6 | **97.2** | **94.8** | **97.0** |

LA-Cond achieves consistent gains across all tasks, with no instances of negative transfer in our evaluation. On average, it achieves a 74.5% success rate under joint training, improving over the Baseline by +20.9%, which corresponds to a +31.8% relative gain compared to its single-task training. As detailed in Appendix E.3, LA-Cond is competitive with several strong baselines. These results suggest that latent action tokens support a consistent modeling formulation for VLA policies under different task distributions.

**Ablation analysis on architectural design.** We examine whether the gains arise from latent action integration or from increased sequence capacity (i.e., longer action placeholders). We construct two baseline variants with placeholder lengths matched to LA-Direct and LA-Cond, denoted as Baseline (PH-Direct) and Baseline (PH-Cond). As shown in Fig. 6, these baselines yield only marginal changes in average success rates (+0.9% for LA-Cond and -1.2% for LA-Direct), indicating that longer placeholders alone do not explain the gains. On long-horizon tasks, the gap remains substantial: on LIBERO-Long, LA-Direct and LA-Cond outperform their matched PH-L baselines by +10.8% and +5.8%, respectively.

Second, we study the regularization depth of LA-Align and apply it to the mid-to-late layers of the VLM by default. Tab. 5 supports this design: aligning at early layers hurts performance (-1.5%) due to insufficient semantic abstraction, while aligning at the final layer is also suboptimal, likely because it is specialized for low-level control.

**Data efficiency in low-resource regimes.** As shown in Appendix E.1, latent action supervision improves sample efficiency under limited data. On LIBERO-Long, LA-Tok achieves 94.0% success with 50% data, outperforming the baseline by +14.0%.

**Sensitivity to latent supervision weight $\lambda$.** We provide a sensitivity study of the latent supervision weight $\lambda$ in Appendix E.4. The results show that performance remains stable across different values of $\lambda$.

## 6. Limitations

Our work has several limitations. First, the integration strategies studied here are abstracted from mapping directions and representative prior designs, and may not cover all possible latent action supervision formulations. Second, although we adopt competent latent action models, it remains open whether stronger latent representations would amplify or narrow the observed gaps between strategies. Finally, our real-world evaluation is limited to a single-arm robotic platform, due to experimental scale and platform availability. Validating these findings across more diverse robot embodiments and environments is an important direction for future work.

## 7. Conclusion

This paper shows that latent actions serve as an effective intermediate representation for VLA training: image-based latent actions benefit long-horizon reasoning and scene-level generalization, action-based latent actions support complex tasks, and directly predicting discrete latent actions achieves the most effective performance, offering practical guidance on how to leverage latent action supervision when training generalist VLA policies.

## Acknowledgements

This work is supported by the National Key Research & Development Plan (2023YFF0725100) and the National Natural Science Foundation of China (92570121, 62322214, U23A20299, U24B20144).

## Impact Statement

This work aims to advance vision-language-action models by systematically studying latent action supervision for robot policy learning. The results may benefit the development of more capable and data-efficient robotic manipulation systems in research and practical settings. We do not identify specific harmful applications beyond the general risks associated with more capable embodied AI systems.

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

# A. Latent Action Model Details

We fine-tune the latent action models on the same training datasets used for policy learning, and freeze them during VLA training.

## A.1. Image-based Latent Action Model

For the image-based latent action model, we adopt the UniVLA architecture (Bu et al., 2025a), a two-stage latent action model pre-trained on OXE. UniVLA uses a frozen DINOv2 visual encoder and a Transformer-based VQ-VAE to model visual transitions between observations (Fig. 7). It partitions the latents into controllable ($z^{\text{ctrl}}$) and uncontrollable ($z^{\text{env}}$) components, and optimize a standard VQ-VAE objective:

$$\mathcal{L}_{\text{img}} = \mathcal{L}_{\text{rec}}^{\text{img}} + \mathcal{L}_{\text{codebook}}^{\text{img}} + \beta \mathcal{L}_{\text{commit}}^{\text{img}}. \tag{11}$$

The reconstruction loss enforces accurate prediction of future visual observations ($o_{t+\delta}$):

$$\mathcal{L}_{\text{rec}}^{\text{img}} = \left\| \hat{o}_{t+\delta} - o_{t+\delta}^* \right\|_2^2, \qquad \delta = 32, \tag{12}$$

where $\hat{o}_{t+\delta}$ is the reconstructed observation at time $t + \delta$, and $o_{t+\delta}^*$ is the corresponding ground-truth observation.

Let $c_t^{\text{img}}$ denote the encoder output before quantization, and let $z_t^{\text{img}}$ the corresponding discrete latent code, with $e(z_t^{\text{img}})$ denoting the selected codebook embedding. We use the standard VQ-VAE codebook and commitment losses:

$$\mathcal{L}_{\text{codebook}}^{\text{img}} = \left\| \text{sg}[c_t^{\text{img}}] - e(z_t^{\text{img}}) \right\|_2^2, \qquad \mathcal{L}_{\text{commit}}^{\text{img}} = \left\| c_t^{\text{img}} - \text{sg}[e(z_t^{\text{img}})] \right\|_2^2, \tag{13}$$

where $\text{sg}[\cdot]$ denotes the stop-gradient operator.

**Action-supervised latent regularization.** In addition to the base UniVLA objective, we incorporate a lightweight auxiliary regularization that leverages a small amount of action supervision during latent action learning.

Recent work has shown that injecting a small amount of ground-truth action supervision during latent action learning can substantially improve the quality and controllability of learned latent actions (Nikulin et al., 2025; Zhang et al., 2025a). Following this insight, we augment the image-based latent action model with a lightweight action prediction head during training (Fig. 7).

Specifically, we attach a multi-layer perceptron (MLP) on top of the pre-quantization latent representation $c_t^{\text{img}}$, which predicts the corresponding low-level action. For a randomly sampled subset of 5% of the training data with available ground-truth actions $a_t^*$, we optimize an additional action regression loss:

$$\mathcal{L}_{\text{act}}^{\text{img}} = \| \hat{a}_t - a_t^* \|_2^2. \tag{14}$$

The full training objective of the image-based latent action model becomes:

$$\mathcal{L} = \mathcal{L}_{\text{img}} + \lambda_{\text{act}} \mathcal{L}_{\text{act}}^{\text{img}}, \tag{15}$$

where $\lambda_{\text{act}}$ balances the auxiliary action supervision, we use $\lambda_{\text{act}} = 1.0$.

To verify the effectiveness of this design choice, we retrain the image-based latent action model without action supervision and evaluate the three image-based strategies on LIBERO. As shown in Tab. 6, all three variants still outperform the baseline (93.1% AVG), and the relative ordering among strategies is preserved; however, a consistent performance drop is observed across all variants compared to the supervised counterpart (Tab. 1), confirming that the 5% action supervision improves latent quality and leads to better downstream policy performance.

Importantly, the action prediction head is used only during latent action learning. During downstream policy learning, the image-based latent action model is frozen and the action head is discarded, ensuring that no ground-truth action information is available at inference time.

## A.2. Action-based Latent Action Model

Our action-based latent action model is a VQ-VAE model that discretizes an action chunk $a_t \in \mathbb{R}^{H \times m}$ into discrete latent tokens $z_t^{\text{act}}$ and reconstructs the original chunk. We use an EMA-updated VQ codebook following VQ-VAE-EMA.

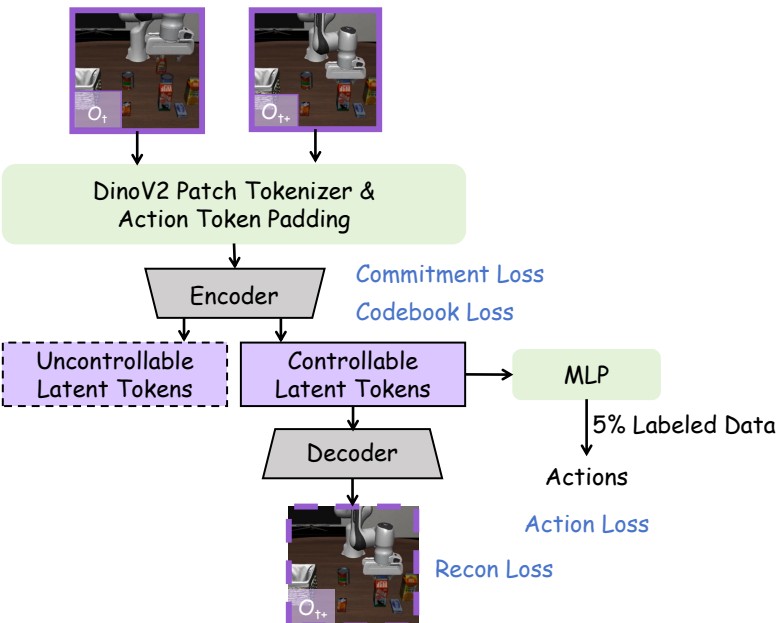

*Figure 7.* Architecture of the fine-tuned image-based latent action model.

*Table 6.* LIBERO results using a vision-only image-based latent action model (without 5% action supervision).

| Method | Spatial | Object | Goal | Long | AVG |
|---|---|---|---|---|---|
| Baseline | 96.6 | 97.2 | 92.8 | 85.8 | 93.1 |
| LA-Align | 97.4 | 98.4 | **97.4** | 94.0 | 96.8 |
| LA-Direct | 97.2 | 98.4 | 97.2 | **94.8** | **96.9** |
| LA-Cond | **98.4** | **99.4** | 96.2 | 92.8 | 96.7 |

**Codebook and Tokenization Interface.** We use a codebook of size $K = 256$ with embedding dimension $d = 128$. The latent action model operates on action chunk with horizon $H$ ($H = 8$ for LIBERO, $H = 25$ for RoboTwin, and $H = 20$ for the real-world Jaka experiment), producing one discrete token per timestep (thus $H$ tokens per chunk). For variable-length trajectories, when the sampled action window is shorter than $H$, we pad by repeating the last action; otherwise, we truncate to the first $H$ steps.

**Architecture.** The encoder augments the action sequence with temporal frequency features (FFT by default, concatenating real and imaginary parts), projects the augmented input to dimension $d = 128$, and applies three 1D temporal convolution layers (kernel size 3, dilation rates $\{1, 2, 4\}$) followed by a 2-layer Transformer encoder (4 heads, feedforward dimension 256). The decoder is a lightweight MLP ($128 \rightarrow 256 \rightarrow m$) applied per timestep.

**Training Loss.** We train the model with the objective:

$$\mathcal{L}_{\text{act}} = \mathcal{L}_{\text{rec}}^{act} + \lambda_{\text{mask}}\mathcal{L}_{\text{mask}} + \beta\mathcal{L}_{\text{commit}}^{act}, \qquad \mathcal{L}_{\text{rec}}^{act} = \|\hat{a}_t - a_t^*\|_2^2, \tag{16}$$

where $\hat{a}_t$ is the reconstructed action chunk and $a_t^*$ is the ground-truth chunk. For VQ regularization, letting $c_t^{\text{act}}$ denote the encoder output before quantization and $z_q = e(z_t^{\text{act}})$ the corresponding quantized embedding sequence, we use:

$$\mathcal{L}_{\text{commit}}^{\text{act}} = \|\text{sg}[z_q] - c_t^{\text{act}}\|_2^2, \tag{17}$$

with EMA codebook updates. We set $\lambda_{\text{mask}} = 0.1$ and $\beta = 0.25$ in all experiments.

**Latent Consistency via Random Masking.** To encourage stable tokenization under partial latent information, we randomly mask a subset of latent timesteps prior to quantization (mask ratio 0.15), decode the masked latents into

reconstructed actions $\tilde{\mathrm{a}}_t$, re-encode them, and match the latents at masked positions:

$$\mathcal{L}_{\text{mask}} = \sum_{h \in \mathcal{M}} \left\| E_\phi(\tilde{\mathrm{a}}_t)_h - E_\phi(\mathrm{a}_t^*)_h \right\|_2^2, \tag{18}$$

where $\mathcal{M}$ denotes the set of masked timestep indices.

**Optimization.** We train the latent action model with Adam (learning rate $1 \times 10^{-4}$, batch size 128) for 50,000 steps. We train a separate model for each benchmark by mixing all training tasks within that benchmark.

## B. Architecture Details

### B.1. Unified VLA Baseline Details

Our unified VLA baseline is constructed upon the Qwen3-VL-2B. To enable parameter-efficient fine-tuning, we integrate Low-Rank Adaptation (LoRA) (Hu et al., 2022) by applying it to all linear layers of the VLM. We configure LoRA with a rank of $r = 64$ and an alpha of $\alpha = 16$. During training, the original weights of the VLM are kept frozen, and only the LoRA parameters and the action head are updated.

**Input Formulation.** The model adopts a parallel decoding formulation, processing a composite input sequence within a single forward pass. This sequence is constructed to include:

- **Visual Observations ($o$):** The visual input comprises a primary camera image and several wrist-mounted camera images. All images are resized to a uniform resolution of $224 \times 224$ pixels before being passed to the VLM.

- **Language Instruction ($\ell$):** The natural language command is processed and appended with a fixed suffix that prompts for action prediction. For a given action chunk size of $H$, this suffix is formatted as: "Please predict the next $H$ robot actions: [ACTION]...[ACTION]", where [ACTION] is a special token repeated $H$ times.

- **Proprioceptive State ($s$):** When utilized, the robot's proprioceptive data is projected from its original dimension to the VLM's hidden dimension $d_{hidden}$ using a simple MLP.

All methods share the same inference procedure and execute a single forward pass; latent-action supervision does not introduce additional rollout steps or iterative planning at test time.

**Action Head Architecture.** The core of the action prediction mechanism is a custom action head inspired by the design of VLA-Adapter (Wang et al., 2025). This head is responsible for regressing a chunk of $H$ continuous action vectors. It extracts the hidden states corresponding to the [ACTION] placeholder tokens from the final half layers of the VLM's transformer backbone.

**Training Objective.** The model is trained end-to-end by minimizing the L2 loss between the predicted action chunk $\hat{\mathrm{a}} \in \mathbb{R}^{H \times m}$ and the ground-truth action sequence $\mathrm{a}^* \in \mathbb{R}^{H \times m}$, where $m$ is the dimensionality of actions.

### B.2. Configuration Across Strategies

To ensure a controlled architectural comparison, all strategies share the same VLM backbone, action head, and aggregation mechanism. The primary architectural difference lies in how placeholder tokens are allocated and supervised within the input sequence. Tab. 7 summarizes the configurations used by each strategy.

For strategies with longer placeholder sequences (LA-Direct and LA-Cond), we include Baseline(PH-L) variants that match the exact placeholder length without latent supervision (Appendix. E.2), confirming that performance gains do not arise from increased token budget.

The difference in $\lambda$ across strategies reflects the scale of the corresponding latent supervision losses rather than a preference for any particular formulation. Specifically, LA-Align (S1) employs a cosine similarity loss for representation alignment, which is bounded and typically has a smaller numerical magnitude. In contrast, LA-Direct, LA-Cond, and LA-Tok (S2–S4) use token-level cross-entropy losses over discrete latent actions, which generally exhibit larger loss values. Accordingly, we set $\lambda = 1.0$ for S1 and $\lambda = 0.1$ for S2–S4 to normalize the contribution of the latent supervision term across strategies and keep the overall loss scales comparable.

*Table 7.* Placeholder configuration across different strategies.

| Strategy | Latent Placeholders | Action Placeholders | Total Length | $\lambda$ (latent guidance in 4.1) |
|---|---|---|---|---|
| Baseline | 0 | $H$ | $H$ | 0 |
| LA-Align (S1) | 0 | $H$ | $H$ | 1.0 |
| LA-Direct (S2) | $P \times H$ | 0 | $P \times H$ | 0.1 |
| LA-Cond (S3) | $P \times H$ | $H$ | $P \times H + H$ | 0.1 |
| LA-Tok (S4) | $H$ | 0 | $H$ | 0.1 |

# C. VLA Baselines and Evaluation Protocols

## C.1. LIBERO

**Evaluation Protocol.** For LIBERO (Tab. 1), we report task success rates (%) under the standard LIBERO evaluation setting. Each task is evaluated with 50 rollouts, and we report results (*Spatial*, *Object*, *Goal*, *Long*) as well as the overall average across all tasks.

**Prior Baselines.** All results for prior VLA methods in Tab. 1 are taken directly from the corresponding papers; we do not re-implement or re-evaluate these baselines. We include both strong VLAs and representative methods that incorporate latent actions:

- **OpenVLA-OFT** (Kim et al., 2025): an optimized fine-tuning recipe built on OpenVLA that uses parallel decoding, action chunking, and a simple regression objective, serving as a strong high-performing VLA reference (7B).

- $\pi_0$ (Black et al., 2024): a generalist VLA trained with a flow-matching diffusion architecture on large-scale multi-robot data (3B).

- **LAPA** (Ye et al., 2024): trains a VQ-VAE to discretize visual transitions into latent actions, then pretrains a backbone with a latent action head, and finally replaces this head with an action head for fine-tuning on downstream tasks (7B).

- **UniVLA** (Bu et al., 2025a): learns task-centric latent actions from videos in the DINO feature space with language conditioning, and trains an action-supervised latent-to-action decoder to enable cross-embodiment policy learning (7B).

- **ThinkAct** (Huang et al., 2025): trains a multimodal LLM to generate latent visual plans via reinforcement signals, and conditions a downstream action policy on the resulting plan latents for long-horizon execution (7B).

- **GR00T** (Bjorck et al., 2025): adopts a dual-system VLA where a vision-language module provides high-level context and a diffusion Transformer generates motor actions, trained end-to-end on heterogeneous data mixtures (2B).

- $\pi_0$-**FAST** (Pertsch et al., 2025): combines $\pi_0$ with a frequency-space discrete action tokenizer (FAST) based on DCT compression, improving action tokenization for high-frequency and dexterous control (3B).

## C.2. RoboTwin 2.0

**Evaluation Protocol.** For RoboTwin 2.0 (Tab. 2), we report success rates (%) following the official evaluation protocol and settings. All baseline numbers are taken from the public RoboTwin 2.0 leaderboard, and we do not re-run or re-evaluate these models.

**Prior Baselines.** We include representative baselines on the leaderboard:

- **RDT** (Liu et al., 2024): a diffusion Transformer foundation model for bimanual manipulation, designed to model multi-modal action distributions and trained on large-scale multi-robot data with a unified action space.

- $\pi_0$ (Black et al., 2024): a generalist VLA trained with a flow-matching diffusion architecture on large-scale multi-robot data.

- **ACT** (Zhao et al., 2023): an imitation learning policy that performs action chunking with a Transformer sequence model, enabling efficient learning of long-horizon manipulation behaviors from limited demonstrations.

- **DP3** (Ze et al., 2024): a diffusion policy conditioned on compact 3D visual representations from point clouds, designed to improve robustness and generalization in dexterous manipulation.

## D. VLA Training Implementation Details

In this section, we provide the detailed hyperparameters and training configurations for our VLA experiments on the LIBERO and RoboTwin benchmarks, and JAKA robot. For all benchmarks, we perform joint training by mixing data across tasks.

### D.1. Training Hyperparameters

We utilize a consistent training recipe across experiments. For a fair comparison, all methods are trained with the same backbone, same training dataset, and identical optimizer settings, and we match the number of optimizer steps and global batch size across all variants. The key hyperparameters are summarized in Tab. 8.

**Optimization Schedule.** For all experiments, we employ a step-decay learning rate scheduler. The learning rate is decayed by a factor of 0.5 every 10,000 steps.

**Action Chunking.** We adopt action chunking to predict future actions in parallel.

*Table 8.* Training hyperparameters for VLA fine-tuning on LIBERO, RoboTwin, benchmark and JAKA robot.

| Hyperparameter | LIBERO | RoboTwin | JAKA |
|---|---|---|---|
| Base VLM | Qwen3VL (2B) | Qwen3VL (2B) | Qwen3VL (2B) |
| Total Training Steps | 80,000 | 30,000 | 30,000 |
| Batch Size | 128 | 128 | 128 |
| Optimizer | AdamW | AdamW | AdamW |
| Learning Rate | $1 \times 10^{-4}$ | $1 \times 10^{-4}$ | $1 \times 10^{-4}$ |
| LR Scheduler | Step Decay | Step Decay | Step Decay |
| Decay Frequency | Every 10k steps | Every 10k steps | Every 10k steps |
| Decay Factor | 0.5 | 0.5 | 0.5 |
| Action Chunk Size ($H$) | 8 | 25 | 20 |

### D.2. Latent Action Specifications

Here we detail the dimensional configurations of the latent actions used in our method.

**Image-based Latent Actions (Section 3.2.1).** Following the fine-tuned UniVLA architecture (Appendix A.1), each discrete visual goal is represented by a sequence of latent tokens.

- Sequence Length: Each temporal step maps to 4 discrete tokens.

- Embedding Dimension: Each token corresponds to a continuous embedding vector of size $\mathbb{R}^{128}$.

- Codebook Size: The discrete quantization uses a compact codebook size of $K_{img} = 16$.

**Action-based Latent Actions (Section 3.2.2).** As detailed in Appendix A.2, our custom latent action model maps an action chunk to a sequence of discrete tokens.

- Sequence Length: Matches the action chunk size ($H$).

- Embedding Dimension: $\mathbb{R}^{128}$.

- Codebook Size: $K_{act} = 256$.

# E. Additional Experimental Results

## E.1. Data Efficiency Analysis

We assess sample efficiency by training on LIBERO benchmark (Fig. 8). Methods with explicit latent action supervision exhibit a clear performance advantage when data is scarce. On LIBERO-Goal, LA-Tok achieves 86.5% success with only 33% of the training data, surpassing the Baseline by +7.9%. Similarly, on the challenging LIBERO-Long suite, it reaches 94.0% success using half the dataset, exceeding the Baseline by +14.0%. The LIBERO-Spatial and LIBERO-Object task results exhibit trends consistent with those observed for the LIBERO-Goal and LIBERO-Long suites. These results confirm that latent actions act as a compressed supervision signal, significantly reducing the sample complexity required to learn effective policies.

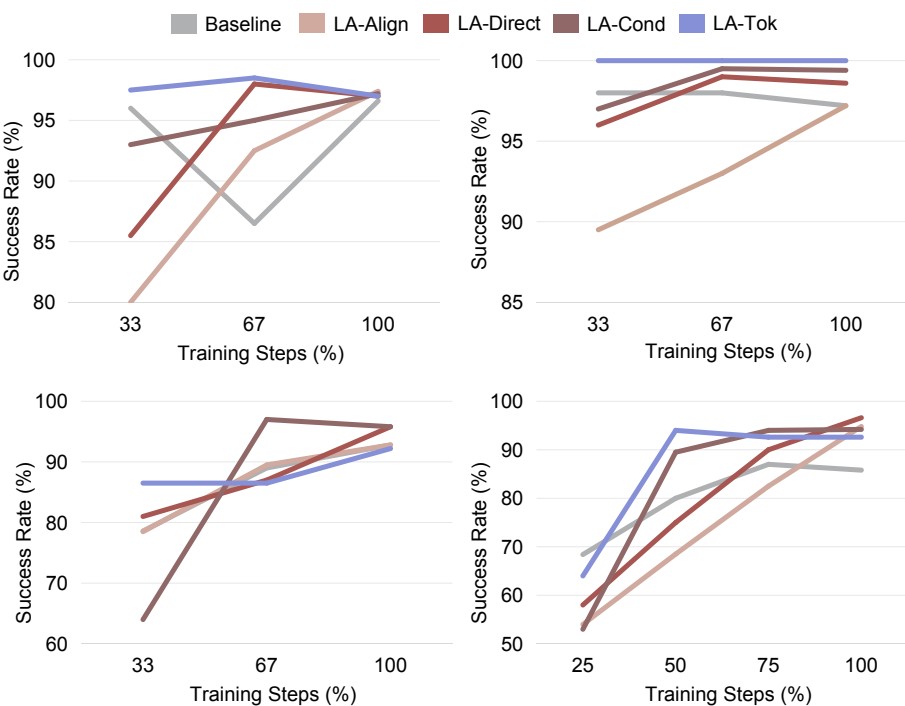

*Figure 8.* Learning curves on LIBERO-Spatial (top left), LIBERO-Object (top right), LIBERO-Goal (bottom left), LIBERO-Long (bottom right).

## E.2. Detailed Ablation on Action Placeholder Length

To verify that performance gains stem from structured latent supervision rather than increased placeholder sequences, we introduce extended placeholder baselines, denoted as Baseline (PH-Direct) and Baseline (PH-Cond). In these baselines, we pad the action placeholder sequences to match the exact token lengths used by LA-Direct and LA-Cond, without altering the training objective. We match the action placeholder length of each strategy separately (32 for LA-Direct and 40 for LA-Cond on LIBERO) to isolate the effect of token budget.

As shown in Tab. 9 and Tab.10, simply allocating more tokens for action prediction yields negligible improvements or even degradation (e.g., -1.2% average drop on LA-Direct). In contrast, both latent action methods achieve significant gains with the same sequence length, particularly on LIBERO-Long. This supports that the performance improvements are driven by latent action supervision rather than increased VLM compute from longer placeholder sequences.

## E.3. Multi-task Joint Training on RoboTwin 2.0.

We evaluate our method on 10 tasks from RoboTwin 2.0. To study the effect of training strategies under heterogeneous task distributions, we consider two protocols: (i) single-task training, where a separate model is trained for each task using only its dataset, and (ii) multi-task joint training, where a single model is trained on the mixture of all 10 task datasets. We report results for both the Baseline and LA-Cond under these two settings. In addition, we include Qwen3OFT (starVLA

*Table 9.* Ablation on action placeholder length ablation for LA-Direct on LIBERO benchmarks.

| Method | Spatial | Object | Goal | Long | AVG |
|---|---|---|---|---|---|
| Baseline | 96.6 | 97.2 | 92.8 | 85.8 | 93.1 |
| Baseline(PH-Direct) | 94.0 | 94.8 | 93.0 | 85.8 | 91.9 |
| LA-Direct | **97.2** | **98.6** | **95.8** | **96.6** | **97.1** |

*Table 10.* Ablation on action placeholder length ablation for LA-Cond on LIBERO benchmarks.

| Method | Spatial | Object | Goal | Long | AVG |
|---|---|---|---|---|---|
| Baseline | 96.6 | 97.2 | 92.8 | 85.8 | 93.1 |
| Baseline(PH-Cond) | 96.8 | 98.6 | 92.0 | 88.4 | 94.0 |
| LA-Cond | **97.0** | **99.4** | **95.8** | **94.2** | **96.6** |

Contributors, 2025) as a multi-task baseline and report its performance under joint training only.

For both training protocols, we follow the same model architecture, optimization setup, and hyperparameters as detailed in Appendix D. The only difference lies in the number of training steps. Specifically, under the single-task training protocol, we train a separate model for each task for 8,000 steps using only task-specific data. Under the multi-task joint training protocol, we train a single model on the combined dataset of all 10 tasks for a total of 60,000 steps.

We first observe that the Baseline suffers from clear negative transfer under joint training: while the average success rate improves from 35.4% to 53.6%, several tasks exhibit substantial performance drops, such as move can pot (-31%) and move playingcard away (-11%). This indicates that naively scaling VLA policies to heterogeneous task mixtures can lead to instability and task interference. In contrast, LA-Cond demonstrates consistent improvements across all tasks when moving from single-task to multi-task joint training. Notably, no task experiences performance degradation.

We further note that under joint training, LA-Cond is competitive with or outperforms several strong multi-task baselines, including RDT, $\pi0$, ACT, and DP-based methods, and closely matches the performance of DP3. Although Qwen3OFT achieves strong results on certain tasks, its performance varies significantly across the task set, whereas LA-Cond exhibits more uniform gains.

| Task | Baseline | | LA-Cond | | Qwen3OFT | RDT | $\pi0$ | ACT | DP | DP3 |
|---|---|---|---|---|---|---|---|---|---|---|
| | Single | Multi | Single | Multi | | | | | | |
| beat block hammer | 12 | 76 | 47 | **85** | 58 | 77 | 43 | 56 | 42 | 72 |
| click bell | 39 | 44 | 33 | 78 | **94** | 80 | 44 | 58 | 54 | 90 |
| grab roller | 48 | 92 | 42 | 97 | 93 | 74 | 96 | 94 | **98** | **98** |
| lift pot | 72 | 91 | 60 | 94 | 0 | 72 | 84 | 88 | 39 | **97** |
| move can pot | 44 | 13 (↓ 31%) | 32 | 63 | 50 | 25 | 58 | 22 | 39 | **70** |
| move playingcard away | 60 | 49 (↓ 11%) | 69 | **80** | 69 | 43 | 53 | 36 | 47 | 68 |
| pick dual bottles | 34 | 34 | 38 | 54 | 43 | 42 | 57 | 31 | 24 | **60** |
| pick diverse bottles | 14 | 40 | 10 | 53 | 30 | 2 | 27 | 7 | 6 | **52** |
| place container plate | 23 | 83 | 70 | 93 | **99** | 78 | 88 | 72 | 41 | 86 |
| place a2b left | 8 | 14 | 26 | **48** | 43 | 3 | 31 | 1 | 2 | 46 |
| AVG | 35.4 | 53.60 | 42.70 | **74.50** | 57.90 | 49.60 | 58.10 | 46.50 | 39.20 | 73.90 |

*Table 11.* Single-task training vs. multi-task joint training on 10 RoboTwin2.0 tasks. Single denotes per-task training, and Multi denotes joint training on all tasks.

### E.4. Sensitivity to Latent Supervision Weight

We analyze the sensitivity of latent action supervision to the weighting coefficient $\lambda$ in the unified objective. We use LA-Tok as a representative strategy for this study and evaluate $\lambda \in \{0.1, 0.2, 0.5\}$ on RoboTwin 2.0, keeping all other training and evaluation settings unchanged.

As shown in Tab. 12, LA-Tok achieves similar performance across different values of $\lambda$. This indicates that the performance of latent action supervision is relatively stable with respect to $\lambda$.

*Table 12.* Sensitivity of LA-Tok to latent supervision weight $\lambda$ on RoboTwin 2.0.

| $\lambda$ | 0.1 | 0.2 | 0.5 |
|---|---|---|---|
| LA-Tok (Avg.) | **78.0** | 76.0 | 76.3 |

## F. Real-World Experiments on the JAKA Arm

To validate the effectiveness of our proposed method in a physical environment, we conducted a series of experiments on a 7-DOF JAKA minicobo robotic arm. The setup includes one static cameras providing third-person perspectives and one wrist-mounted camera for a first-person view, as illustrated in Fig. 9, 10 and 11.

Our real-world evaluation covers both manipulation tasks and cluttered tabletop pick-and-place tasks, allowing us to examine long-horizon reasoning, contact-rich manipulation, and scene-level generalization.

### F.1. Task Descriptions and Data Collection

We design two groups of real-world tasks: manipulation tasks and pick-and-place tasks. Each task corresponds to a high-level language instruction. For each task, we collect 50 demonstrations.

- **Manipulation Task 1: Stack the Bowls.** The command for this task is *"stack the bowls"*. This task is further divided into three variations involving stacking two, three, or four bowls, with 20, 15, and 15 demonstrations collected for each setting, respectively (50 demonstrations total). A key challenge of this task is its requirement for sequential manipulation. For example, to stack four bowls, the policy must first create a two-bowl stack, then grasp this stack to place it onto a third bowl, and finally grasp the resulting three-bowl stack to place it onto the fourth. The initial positions of the bowls are randomized to test the policy's adaptability. The visual setup is shown in Fig. 9.

- **Manipulation Task 2: Wipe off the Stain.** The command for this task is *"wipe the dark stain from the table"*. This is a two-stage task requiring the robot to first locate and grasp a cleaning cloth from the table, and then use it to wipe a dark, liquid stain (simulated with spilled cola). To ensure robustness, the color and initial position of the cleaning cloth were varied across the 50 demonstrations. The visual setup is shown in Fig. 10.

- **Pick-and-Place Tasks.** We additionally design three cluttered tabletop pick-and-place tasks targeting different objects: mango, sponge, and bottle (the mango task involves an irregular object, while the sponge task involves a deformable and contact-rich object). In each task, the robot is instructed to pick up the target object and place it at the designated location. For each pick-and-place task, we collect 50 demonstrations. The scenes include distractor objects in addition to the target object, enabling evaluation of robustness to clutter and scene variation. The visual setup is shown in Fig. 11.

### F.2. Evaluation Protocol

**Evaluation setup.** To quantitatively assess real-world performance, we evaluated our baseline and four variations. For each task, every model was evaluated for 10 independent rollouts. For manipulation tasks, rollouts are executed from randomized initial object placements within a predefined workspace region, while keeping the camera viewpoint and robot calibration fixed throughout the evaluation. For pick-and-place tasks, we evaluate under both in-domain (ID) and out-of-domain (OOD) settings. In the ID setting, object placements and scene layouts are similar to those in the training demonstrations. In the OOD setting, we use unseen scene configurations by rearranging the tabletop layout and introducing

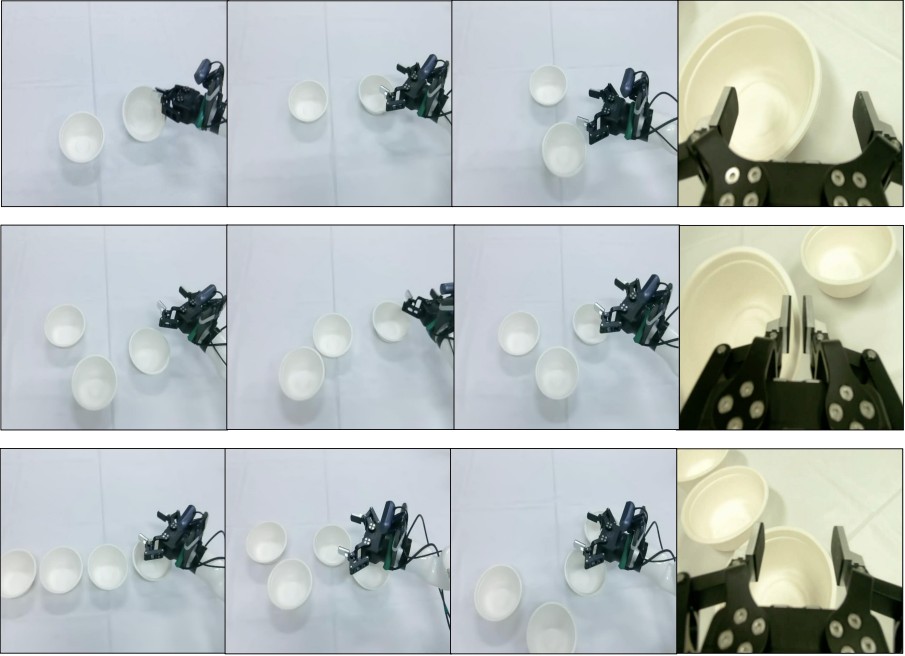

*Figure 9.* Visualizations of stack the bowls tasks.

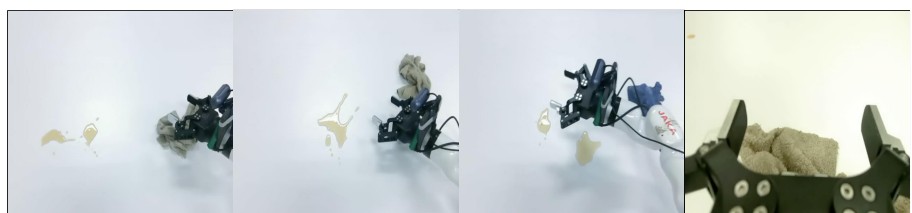

*Figure 10.* Visualizations of wipe off the stain task.

distractor objects not observed during training. An evaluation episode was terminated either when the task was successfully completed or when no correct action was executed for 20 seconds, whichever occurred first.

**Success metric.** We use different success metrics for manipulation and pick-and-place tasks. For manipulation tasks, we report a normalized completion score $S \in [0, 1]$, which captures partial completion and is more informative than a binary success indicator in real-world manipulation. For readability, we present the score as Score $= 100 \times S$ in the main paper. For pick-and-place tasks, success is measured as binary task completion: a rollout is successful if the robot successfully picks up the target object and places it at the designated location. We report the binary success rate over 10 rollouts, also scaled to $[0, 100]$.

- **Stacking tasks.** Success is measured by the degree of proper nesting for each placed bowl. We use **paper bowls**, which have relatively high friction and therefore do not automatically slide into a fully nested configuration. As a result, partially nested outcomes are frequent, and a graded metric is more informative than a binary success definition.

  For each bowl placement, we assign an individual nesting score $s \in [0, 1]$ using the following rubric based on the final bowl configuration:

  - $s = 0.0$: the stacking action is not successfully completed within 20 seconds, i.e., the bowl is not inserted into the target bowl by the end of the rollout.
  - $s = 0.5$: the bowl is inserted but remains largely vertical/unstable (e.g., "standing" inside the target bowl without being seated).

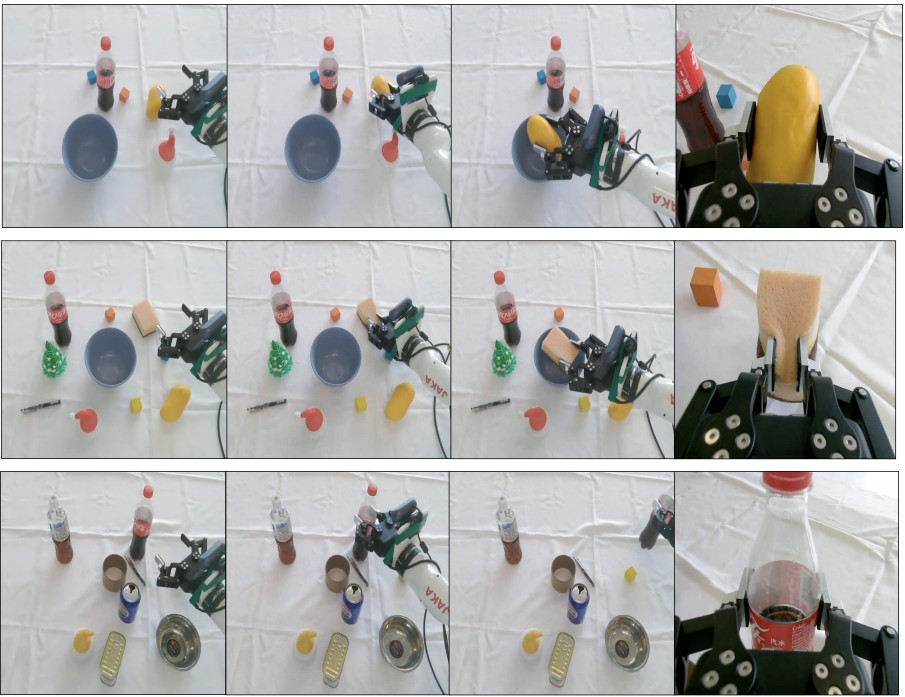

*Figure 11.* Visualizations of pick-and-place task.

- $s = 0.8$: the bowl is mostly nested and seated, but not perfectly stacked (small yet visible gap/misalignment remains).
- $s = 1.0$: the bowl is fully nested and stably seated, matching the intended stacked configuration.

For tasks involving multiple bowls, the final stacking score is computed as the average of $N$ individual bowl scores:

$$S_{\text{stack}} = \frac{1}{N} \sum_{i=1}^{N} s_i.$$

- **Wiping task.** The wiping score is determined by the percentage of the stain's surface area that is successfully cleaned by the cloth. Concretely, we define the score as:

$$S_{\text{wipe}} = 1 - \frac{A_{\text{remain}}}{A_{\text{init}}},$$

where $A_{\text{init}}$ is the stain area at the beginning of the rollout and $A_{\text{remain}}$ is the remaining stain area at the end. The score is clipped to $[0, 1]$.

- **Pick-and-place tasks.** For pick-and-place tasks, each rollout is evaluated by two sub-goals: whether the robot successfully picks up the target object and whether it places the object at the designated location. Each sub-goal contributes 0.5 to the rollout score, resulting in a score of 1.0 if both picking and placing are completed, 0.5 if only one sub-goal is completed, and 0.0 otherwise. For each task and setting, we report the average score over 10 rollouts, scaled to a $[0, 100]$ range in the main paper.

**Aggregating scores.** For each model and each task variation, we report the mean score across rollouts:

$$\bar{S} = \frac{1}{R} \sum_{r=1}^{R} S^{(r)},$$

where $R = 10$ denotes the number of independent rollouts.

### F.3. Quantitative results.

*Table 13.* Baseline results on stacking and wiping tasks.

| Baseline | Stack 2 bowls sucess | Stack 3 bowls | | | Stack 4 bowls | | | | Wipe off stains sucess |
|---|---|---|---|---|---|---|---|---|---|
| | | sucess 2 | sucess 3 | AVG | sucess 2 | sucess 3 | sucess 4 | AVG | |
| 1 | 0.9 | 1.0 | 0.6 | 0.8 | 1.0 | 0.5 | 0.0 | 0.50 | 1.00 |
| 2 | 0.7 | 0.7 | 0.0 | 0.35 | 0.0 | 0.0 | 0.0 | 0.00 | 0.90 |
| 3 | 0.8 | 1.0 | 0.4 | 0.7 | 1.0 | 1.0 | 0.0 | 0.67 | 0.95 |
| 4 | 0.7 | 1.0 | 0.8 | 0.9 | 1.0 | 1.0 | 0.0 | 0.67 | 0.90 |
| 5 | 0.6 | 0.7 | 0.0 | 0.35 | 1.0 | 0.8 | 0.0 | 0.60 | 0.80 |
| 6 | 0.8 | 0.9 | 0.6 | 0.75 | 0.9 | 0.7 | 0.0 | 0.53 | 0.80 |
| 7 | 0.8 | 1.0 | 0.8 | 0.90 | 0.7 | 0.0 | 0.0 | 0.23 | 0.70 |
| 8 | 0.7 | 0.7 | 0.0 | 0.35 | 0.8 | 0.6 | 0.0 | 0.47 | 0.90 |
| 9 | 1.0 | 0.7 | 0.5 | 0.60 | 0.9 | 0.6 | 0.0 | 0.50 | 1.00 |
| 10 | 0.7 | 0.8 | 0.5 | 0.65 | 1.0 | 0.8 | 0.0 | 0.60 | 0.90 |
| AVG | 0.77 | 0.64 | | | 0.48 | | | | 0.89 |

*Table 14.* LA-Align results on stacking and wiping tasks.

| LA-Align | Stack 2 bowls sucess | Stack 3 bowls | | | Stack 4 bowls | | | | Wipe off stains sucess |
|---|---|---|---|---|---|---|---|---|---|
| | | sucess 2 | sucess 3 | AVG | sucess 2 | sucess 3 | sucess 4 | AVG | |
| 1 | 0.9 | 0.8 | 0.7 | 0.75 | 1.0 | 0.6 | 0.0 | 0.53 | 0.80 |
| 2 | 0.8 | 0.9 | 0.7 | 0.80 | 0.8 | 0.0 | 0.0 | 0.27 | 0.90 |
| 3 | 1.0 | 0.9 | 0.6 | 0.75 | 0.9 | 0.0 | 0.0 | 0.30 | 1.00 |
| 4 | 0.9 | 0.9 | 0.9 | 0.90 | 0.8 | 1.0 | 0.5 | 0.77 | 1.00 |
| 5 | 0.8 | 0.7 | 0.5 | 0.60 | 0.8 | 0.8 | 0.4 | 0.67 | 1.00 |
| 6 | 0.7 | 0.7 | 0.7 | 0.70 | 0.8 | 0.7 | 0.0 | 0.50 | 0.90 |
| 7 | 0.8 | 1.0 | 0.8 | 0.90 | 1.0 | 0.7 | 0.0 | 0.57 | 0.80 |
| 8 | 1.0 | 0.8 | 0.7 | 0.75 | 0.8 | 0.5 | 0.0 | 0.43 | 0.90 |
| 9 | 0.7 | 0.8 | 0.7 | 0.75 | 0.9 | 0.6 | 0.0 | 0.50 | 0.90 |
| 10 | 0.9 | 0.7 | 0.6 | 0.65 | 0.8 | 0.8 | 0.6 | 0.73 | 1.00 |
| AVG | 0.85 | 0.76 | | | 0.53 | | | | 0.92 |

*Table 15.* LA-Direct results on stacking and wiping tasks.

| LA-Direct | Stack 2 bowls sucess | Stack 3 bowls | | | Stack 4 bowls | | | | Wipe off stains sucess |
|---|---|---|---|---|---|---|---|---|---|
| | | sucess 2 | sucess 3 | AVG | sucess 2 | sucess 3 | sucess 4 | AVG | |
| 1 | 0.9 | 0.7 | 1.0 | 0.85 | 1.0 | 0.8 | 0.6 | 0.80 | 0.95 |
| 2 | 0.8 | 1.0 | 0.8 | 0.90 | 1.0 | 1.0 | 0.0 | 0.67 | 1.00 |
| 3 | 0.7 | 1.0 | 0.8 | 0.90 | 1.0 | 0.8 | 0.8 | 0.87 | 1.00 |
| 4 | 0.7 | 1.0 | 1.0 | 1.00 | 1.0 | 1.0 | 0.0 | 0.67 | 1.00 |
| 5 | 0.8 | 1.0 | 0.8 | 0.90 | 1.0 | 1.0 | 1.0 | 1.00 | 1.00 |
| 6 | 0.8 | 1.0 | 0.8 | 0.90 | 0.8 | 1.0 | 0.7 | 0.83 | 0.90 |
| 7 | 1.0 | 0.9 | 0.7 | 0.80 | 1.0 | 0.8 | 0.0 | 0.60 | 0.95 |
| 8 | 1.0 | 0.9 | 0.8 | 0.85 | 1.0 | 0.8 | 0.5 | 0.77 | 0.95 |
| 9 | 0.7 | 0.8 | 0.8 | 0.80 | 1.0 | 1.0 | 0.5 | 0.83 | 1.00 |
| 10 | 0.9 | 1.0 | 0.8 | 0.90 | 1.0 | 1.0 | 0.7 | 0.90 | 1.00 |
| AVG | 0.83 | 0.88 | | | 0.79 | | | | 0.98 |

*Table 16.* LA-Cond results on stacking and wiping tasks.

| LA-Cond | Stack 2 bowls sucess | Stack 3 bowls sucess 2 | sucess 3 | AVG | Stack 4 bowls sucess 2 | sucess 3 | sucess 4 | AVG | Wipe off stains sucess |
|---------|------|------|------|------|------|------|------|------|------|
| 1  | 0.8 | 1.0 | 0.0 | 0.50 | 1.0 | 1.0 | 0.0 | 0.67 | 1.00 |
| 2  | 0.8 | 1.0 | 0.0 | 0.50 | 1.0 | 1.0 | 0.8 | 0.93 | 0.95 |
| 3  | 0.9 | 1.0 | 0.6 | 0.80 | 1.0 | 1.0 | 0.0 | 0.67 | 1.00 |
| 4  | 0.8 | 0.8 | 0.5 | 0.65 | 0.8 | 1.0 | 0.0 | 0.60 | 0.80 |
| 5  | 0.7 | 1.0 | 0.8 | 0.90 | 0.8 | 0.7 | 0.0 | 0.50 | 0.85 |
| 6  | 0.8 | 1.0 | 0.8 | 0.90 | 1.0 | 1.0 | 0.8 | 0.93 | 1.00 |
| 7  | 1.0 | 1.0 | 0.5 | 0.75 | 1.0 | 0.6 | 0.6 | 0.73 | 1.00 |
| 8  | 1.0 | 0.8 | 0.6 | 0.70 | 0.8 | 0.7 | 0.5 | 0.67 | 0.90 |
| 9  | 0.8 | 1.0 | 0.0 | 0.50 | 1.0 | 0.6 | 0.0 | 0.53 | 0.85 |
| 10 | 0.6 | 1.0 | 0.6 | 0.80 | 0.8 | 0.5 | 0.0 | 0.43 | 0.80 |
| AVG | 0.82 | | 0.70 | | | 0.67 | | | 0.92 |

*Table 17.* LA-Tok results on stacking and wiping tasks.

| LA-Tok | Stack 2 bowls sucess | Stack 3 bowls sucess 2 | sucess 3 | AVG | Stack 4 bowls sucess 2 | sucess 3 | sucess 4 | AVG | Wipe off stains sucess |
|--------|------|------|------|------|------|------|------|------|------|
| 1  | 1.0 | 0.5 | 1.0 | 0.75 | 1.0 | 1.0 | 0.0 | 0.67 | 0.90 |
| 2  | 1.0 | 1.0 | 1.0 | 1.00 | 0.8 | 0.5 | 0.0 | 0.43 | 1.00 |
| 3  | 0.8 | 0.8 | 0.5 | 0.65 | 0.8 | 0.6 | 0.0 | 0.47 | 1.00 |
| 4  | 1.0 | 0.9 | 1.0 | 0.95 | 1.0 | 1.0 | 1.0 | 1.00 | 0.70 |
| 5  | 0.9 | 0.9 | 0.8 | 0.85 | 1.0 | 0.8 | 0.0 | 0.60 | 0.95 |
| 6  | 0.8 | 0.9 | 0.7 | 0.80 | 1.0 | 0.6 | 0.0 | 0.53 | 1.00 |
| 7  | 0.9 | 1.0 | 1.0 | 1.00 | 0.9 | 0.7 | 0.5 | 0.70 | 1.00 |
| 8  | 1.0 | 1.0 | 0.6 | 0.80 | 0.8 | 0.6 | 0.0 | 0.47 | 0.95 |
| 9  | 1.0 | 0.8 | 0.5 | 0.65 | 1.0 | 1.0 | 0.5 | 0.83 | 0.80 |
| 10 | 0.8 | 0.9 | 0.7 | 0.80 | 1.0 | 0.8 | 0.5 | 0.77 | 0.90 |
| AVG | 0.92 | | 0.83 | | | 0.65 | | | 0.92 |

*Table 18.* Baseline results on pick-and-place tasks.

| Baseline | Rollout | Mango Pick | Place | Score | Sponge Pick | Place | Score | Bottle Score | Avg. Score |
|----------|---------|------|------|------|------|------|------|------|------|
|    | 1  | 1.0 | 1.0 | 1.00 | 1.0 | 0.0 | 0.50 | 1.00 | – |
|    | 2  | 1.0 | 1.0 | 1.00 | 1.0 | 1.0 | 1.00 | 0.50 | – |
| ID | 3  | 1.0 | 1.0 | 1.00 | 1.0 | 0.0 | 0.50 | 0.00 | – |
|    | 4  | 1.0 | 1.0 | 1.00 | 0.0 | 0.0 | 0.00 | 1.00 | – |
|    | 5  | 1.0 | 1.0 | 1.00 | 1.0 | 1.0 | 1.00 | 1.00 | – |
| Avg. (ID) | | 1.0 | 1.0 | 1.00 | 0.8 | 0.4 | 0.60 | 0.70 | 0.77 |
|    | 6  | 1.0 | 0.0 | 0.50 | 0.0 | 0.0 | 0.00 | 0.00 | – |
|    | 7  | 1.0 | 0.0 | 0.50 | 1.0 | 0.0 | 0.50 | 0.00 | – |
| OOD | 8  | 1.0 | 0.0 | 0.50 | 1.0 | 0.0 | 0.50 | 0.00 | – |
|    | 9  | 0.0 | 0.0 | 0.00 | 0.0 | 0.0 | 0.00 | 0.00 | – |
|    | 10 | 0.0 | 0.0 | 0.00 | 0.0 | 0.0 | 0.00 | 0.00 | – |
| Avg. (OOD) | | 0.6 | 0.0 | 0.30 | 0.4 | 0.0 | 0.20 | 0.00 | 0.17 |
| Avg. (Total) | | 0.8 | 0.5 | 0.60 | 0.6 | 0.2 | 0.40 | 0.35 | 0.47 |

*Table 19.* LA-Align results on pick-and-place tasks.

| LA-Align | Rollout | Mango | | | Sponge | | | Bottle | Avg. |
|---|---|---|---|---|---|---|---|---|---|
| | | Pick | Place | Score | Pick | Place | Score | Score | Score |
| | 1 | 1.0 | 1.0 | 1.00 | 1.0 | 1.0 | 1.00 | 1.00 | – |
| | 2 | 1.0 | 1.0 | 1.00 | 1.0 | 1.0 | 1.00 | 0.50 | – |
| ID | 3 | 1.0 | 1.0 | 1.00 | 1.0 | 1.0 | 1.00 | 1.00 | – |
| | 4 | 1.0 | 0.0 | 0.50 | 1.0 | 0.0 | 0.50 | 0.00 | – |
| | 5 | 1.0 | 1.0 | 1.00 | 1.0 | 1.0 | 1.00 | 0.50 | – |
| Avg. (ID) | | 1.0 | 0.8 | 0.90 | 1.0 | 0.8 | 0.90 | 0.60 | 0.80 |
| | 6 | 1.0 | 1.0 | 1.00 | 1.0 | 0.0 | 0.50 | 0.50 | – |
| | 7 | 0.0 | 0.0 | 0.00 | 1.0 | 1.0 | 1.00 | 0.00 | – |
| OOD | 8 | 1.0 | 1.0 | 1.00 | 1.0 | 1.0 | 1.00 | 0.50 | – |
| | 9 | 1.0 | 1.0 | 1.00 | 0.0 | 0.0 | 0.00 | 0.00 | – |
| | 10 | 1.0 | 1.0 | 1.00 | 1.0 | 0.0 | 0.50 | 0.50 | – |
| Avg. (OOD) | | 0.8 | 0.8 | 0.80 | 0.8 | 0.4 | 0.60 | 0.30 | 0.57 |
| Avg. (Total) | | 0.9 | 0.8 | 0.85 | 0.9 | 0.6 | 0.75 | 0.45 | 0.68 |

*Table 20.* LA-Direct results on pick-and-place tasks.

| LA-Direct | Rollout | Mango | | | Sponge | | | Bottle | Avg. |
|---|---|---|---|---|---|---|---|---|---|
| | | Pick | Place | Score | Pick | Place | Score | Score | Score |
| | 1 | 1.0 | 1.0 | 1.00 | 1.0 | 1.0 | 1.00 | 0.50 | – |
| | 2 | 1.0 | 1.0 | 1.00 | 1.0 | 1.0 | 1.00 | 0.50 | – |
| ID | 3 | 1.0 | 1.0 | 1.00 | 1.0 | 1.0 | 1.00 | 1.00 | – |
| | 4 | 1.0 | 1.0 | 1.00 | 1.0 | 1.0 | 1.00 | 0.00 | – |
| | 5 | 1.0 | 1.0 | 1.00 | 1.0 | 0.0 | 0.50 | 1.00 | – |
| Avg. (ID) | | 1.0 | 1.0 | 1.00 | 1.0 | 0.8 | 0.90 | 0.60 | 0.83 |
| | 6 | 0.0 | 0.0 | 0.00 | 1.0 | 1.0 | 1.00 | 0.50 | – |
| | 7 | 1.0 | 1.0 | 1.00 | 1.0 | 0.0 | 0.50 | 0.00 | – |
| OOD | 8 | 1.0 | 0.0 | 0.50 | 1.0 | 0.0 | 0.50 | 0.50 | – |
| | 9 | 1.0 | 1.0 | 1.00 | 1.0 | 1.0 | 1.00 | 1.00 | – |
| | 10 | 1.0 | 1.0 | 1.00 | 1.0 | 1.0 | 1.00 | 0.50 | – |
| Avg. (OOD) | | 0.8 | 0.6 | 0.70 | 1.0 | 0.6 | 0.80 | 0.50 | 0.67 |
| Avg. (Total) | | 0.9 | 0.8 | 0.85 | 1.0 | 0.7 | 0.85 | 0.55 | 0.75 |

*Table 21.* LA-Cond results on pick-and-place tasks.

| LA-Cond | Rollout | Mango | | | Sponge | | | Bottle | Avg. |
|---|---|---|---|---|---|---|---|---|---|
| | | Pick | Place | Score | Pick | Place | Score | Score | Score |
| | 1 | 1.0 | 1.0 | 1.00 | 1.0 | 1.0 | 1.00 | 0.00 | – |
| | 2 | 1.0 | 1.0 | 1.00 | 1.0 | 1.0 | 1.00 | 1.00 | – |
| ID | 3 | 1.0 | 1.0 | 1.00 | 1.0 | 1.0 | 1.00 | 1.00 | – |
| | 4 | 1.0 | 1.0 | 1.00 | 1.0 | 0.0 | 0.50 | 0.50 | – |
| | 5 | 1.0 | 0.0 | 0.50 | 1.0 | 1.0 | 1.00 | 0.00 | – |
| Avg. (ID) | | 1.0 | 0.8 | 0.90 | 1.0 | 0.8 | 0.90 | 0.50 | 0.77 |
| | 6 | 1.0 | 1.0 | 1.00 | 1.0 | 1.0 | 1.00 | 0.50 | – |
| | 7 | 1.0 | 1.0 | 1.00 | 1.0 | 1.0 | 1.00 | 0.00 | – |
| OOD | 8 | 1.0 | 0.0 | 0.50 | 1.0 | 1.0 | 1.00 | 0.50 | – |
| | 9 | 1.0 | 1.0 | 1.00 | 0.0 | 0.0 | 0.00 | 1.00 | – |
| | 10 | 1.0 | 1.0 | 1.00 | 1.0 | 1.0 | 1.00 | 0.00 | – |
| Avg. (OOD) | | 1.0 | 0.8 | 0.90 | 0.8 | 0.8 | 0.80 | 0.40 | 0.70 |
| Avg. (Total) | | 1.0 | 0.8 | 0.90 | 0.9 | 0.8 | 0.85 | 0.45 | 0.73 |

*Table 22.* LA-Tok results on pick-and-place tasks.

| LA-Tok | Rollout | Mango | | | Sponge | | | Bottle | Avg. |
|---|---|---|---|---|---|---|---|---|---|
| | | Pick | Place | Score | Pick | Place | Score | Score | Score |
| | 1 | 1.0 | 1.0 | 1.00 | 1.0 | 1.0 | 1.00 | 0.50 | – |
| | 2 | 1.0 | 1.0 | 1.00 | 1.0 | 1.0 | 1.00 | 0.00 | – |
| ID | 3 | 1.0 | 1.0 | 1.00 | 1.0 | 1.0 | 1.00 | 1.00 | – |
| | 4 | 1.0 | 1.0 | 1.00 | 1.0 | 0.0 | 0.50 | 0.50 | – |
| | 5 | 1.0 | 1.0 | 1.00 | 1.0 | 1.0 | 1.00 | 1.00 | – |
| Avg. (ID) | | 1.0 | 1.0 | 1.00 | 1.0 | 0.8 | 0.90 | 0.60 | 0.83 |
| | 6 | 1.0 | 0.0 | 0.50 | 1.0 | 0.0 | 0.50 | 0.00 | – |
| | 7 | 1.0 | 1.0 | 1.00 | 1.0 | 1.0 | 1.00 | 0.50 | – |
| OOD | 8 | 1.0 | 0.0 | 0.50 | 1.0 | 0.0 | 0.50 | 0.00 | – |
| | 9 | 1.0 | 0.0 | 0.50 | 1.0 | 1.0 | 1.00 | 0.50 | – |
| | 10 | 1.0 | 0.0 | 0.50 | 1.0 | 0.0 | 0.50 | 0.00 | – |
| Avg. (OOD) | | 1.0 | 0.2 | 0.60 | 1.0 | 0.4 | 0.70 | 0.20 | 0.50 |
| Avg. (Total) | | 1.0 | 0.6 | 0.80 | 1.0 | 0.6 | 0.80 | 0.40 | 0.67 |

