# OpenReview forum: "From Pixels to Tokens: A Systematic Study of Latent Action Supervision for Vision-Language-Action Models"
_ICML.cc/2026/Conference — ICML 2026 spotlight_

### Official Review · Reviewer_erJi · 2026-03-11

**Soundness:** 3
**Presentation:** 3
**Significance:** 3
**Originality:** 2
**Overall Recommendation:** 5
**Confidence:** 4

**Summary:**

This paper systematically studies how latent action supervision should be integrated into vision-language-action models, comparing image-based and action-based latent formulations under a unified VLA baseline across multiple benchmarks and real-world experiments, and finding that discrete latent token supervision provides consistent performance benefits.

**Compliance With Llm Reviewing Policy:**

Affirmed.

**Final Justification:**

My questions are resolved.

**Key Questions For Authors:**

1. To what extent do the observed differences come from the latent formulation itself, rather than the prior knowledge of the underlying latent models?
2. How sensitive are the main conclusions to the latent supervision weight lambda?
3. Since the image-based latent model incorporates a small amount of action supervision during training, have the authors tested a purely vision-only version to verify whether the main conclusions still hold?
4. Did the authors observe any training instability or convergence differences between the four strategies?
5. Do the authors expect the observed formulation-task correspondence  to generalize to other VLA architectures or larger backbone models?

**Limitations:**

Yes

**Strengths And Weaknesses:**

Strengths:
- The paper addresses an important and timely problem.. A systematic comparison under a unified baseline is therefore valuable.
- The distinction between trajectory regularization and target-space unification provides a useful conceptual framework.

Weaknesses:
- The two latent action models differ substantially in architecture, training setup, and prior knowledge. The image-based latent model is based on UniVLA, while the action-based latent model is a custom tokenizer trained from scratch per benchmark. Moreover, the image-based latent model is augmented with 5% action supervision during latent learning. As a result, some of the reported formulation-task correspondence may stem from differences in the underlying latent models, not purely from the formulation itself.
- Although the paper repeatedly states that it does not compare latent action models themselves, the downstream conclusions still depend on latent model quality. This is important because the claim of the paper is about when image-based versus action-based latent actions should be preferred.
- I did not see std reporting or other analysis for the main simulation results. This is not critical for the larger improvements, but it matters for smaller differences in the paper.

---

> ### Author Rebuttal · Authors · 2026-03-30
>
> We thank the reviewer for the thorough and technically detailed feedback, and the positive assessment of our conceptual framework and systematic comparison. We address each concern in turn.
>
> **Weakness 1 & 2 & Q1**: The formulation-task correspondence may stem from differences in the underlying latent models rather than the formulation itself.
>
> We first clarify that conclusions 2 and 3 are derived from within-formulation comparisons and are unaffected by cross-formulation differences in latent model quality.
>
> For conclusion 1, we note that the difference in model quality actually supports rather than undermines the correspondence. The two latent models differ in what they fundamentally learn: image-based latent models must capture structured visual transitions between frames, a capability that depends critically on large-scale pretraining; action-based latent models, by contrast, learn to discretize actions within a specific action space, which does not require large-scale pretraining for benchmark-level tasks.
>
> If the correspondence were purely an artifact of model quality, image-based latent actions should outperform action-based ones uniformly across all tasks and strategies. However, on RoboTwin 2.0, LA-Tok (78.0%) substantially outperforms LA-Align (70.5%) and also in real-world experiments. These results suggest that the formulation-task correspondence reflects a property of the formulations themselves rather than latent model quality differences.
>
> **Weakness 3**: Standard deviation is not reported for the main simulation results.
>
> The main conclusions of this paper are drawn from results with substantial performance gaps, particularly on LIBERO-Long and RoboTwin 2.0. Real-world experiments further corroborate these trends. Due to the large number of experimental variants and time constraints, we were unable to complete full multi-run evaluations in a week, but we plan to include standard deviation reporting in the final version.
>
> **Q2**: How sensitive are the main conclusions to the latent supervision weight $\lambda$?
>
> The choice of $\lambda$ is designed to normalize the latent supervision loss to a similar numerical scale as the action loss across strategies. To verify robustness, we conducted sensitivity experiments with $\lambda \in \\{0.1, 0.2, 0.5 \\}$ for LA-Tok on RoboTwin 2.0:
>
> | $\lambda$ | 0.1 (paper) | 0.2 | 0.5 |
> |:-------|:----------:|:-----------:|:-----------:|
> | LA-Tok (Avg.) | 78.0 | 76.0 | 76.3 |
>
> Performance remains stable across all settings, with an average drop of less than 2%, confirming that the main conclusions are not sensitive to the choice of $\lambda$.
>
> **Q3**: Has a purely vision-only image-based latent model been tested to verify whether the main conclusions still hold?
>
> The 5% action supervision follows recent work on latent action training quality (Nikulin et al., 2025; Zhang et al., 2025a), and is intended to improve latent quality. We retrained the image-based latent action model without this supervision and evaluated the three image-based variants on LIBERO. The relative ordering among strategies is preserved on this benchmark and especially on LIBERO-Long, confirming the main conclusions hold (all three image-based variants still outperform LA-Tok from the paper). A slight overall performance drop is observed, suggesting the action supervision provides better latent quality.
>
> | Method | Spatial | Object | Goal | Long | AVG |
> |:------|:-------:|:------:|:----:|:----:|:---:|
> | LA-Align (vision-only) | 97.4 | 98.4 | 97.4 | 94.0 | 96.8 |
> | LA-Direct (vision-only) | 97.2 | 98.4 | 97.2 | **94.8** | **96.9** |
> | LA-Cond (vision-only) | 98.4 | 99.4 | 96.2 | 92.8 | 96.7 |
>
> **Q4**: Were any training instability or convergence differences observed between the four strategies?
>
> We observed that LA-Cond converges faster in terms of training loss, which we attribute to its hierarchical structure providing denser gradient signals through both latent and action segments simultaneously. We will include this observation in the revised paper.
>
> **Q5**: Is the observed formulation-task correspondence expected to generalize to other VLA architectures or larger backbone models?
>
> We expect the findings to generalize, as all four strategies operate exclusively at the level of VLM supervision, making them applicable in principle to any VLM backbone such as InternVL or Prismatic VLM. The core findings are grounded in general properties of VLM. Adapting the implementation to different backbones requires non-trivial engineering effort due to differences in formulation; given the 7-day rebuttal window, we were unable to complete these experiments, but plan to include results on additional backbones in the final version.
>
> [1] Nikulin, Alexander, et al. "Latent action learning requires supervision in the presence of distractors." arXiv preprint arXiv:2502.00379 (2025).
>
> [2] Zhang, Chuheng, et al. "What Do Latent Action Models Actually Learn?." arXiv preprint arXiv:2506.15691 (2025).

---

> > ### Author Rebuttal · Reviewer_erJi · 2026-04-03
> >
> > I have no further questions. And I raise my score to 5

---

### Official Review · Reviewer_ngJP · 2026-03-12

**Soundness:** 3
**Presentation:** 3
**Significance:** 3
**Originality:** 2
**Overall Recommendation:** 4
**Confidence:** 3

**Summary:**

This paper presents a systematic study of how latent actions can improve Vision-Language-Action (VLA) models. The paper categorizes latent action supervision into two distinct roles: Regularizing the Trajectory and Unifying the Target Space. Through extensive benchmarking in simulation (LIBERO, RobotWin2.0) and real-world robot experiments (JAKA arm), the paper identifies Action-to-Token Mapping as the most robust integration strategy for building generalist robotic policies.

**Compliance With Llm Reviewing Policy:**

Affirmed.

**Final Justification:**

My concerns have been addressed.

**Key Questions For Authors:**

Do the authors plan to release the unified VLA baseline code and the collected JAKA dataset? Given that the paper’s value lies in its systematic comparison, public access to this standardized framework is essential for the community to build upon these findings.

Can the authors clarify if any specific components are technically novel compared to prior works? A clearer distinction between re-implementing existing ideas and original technical contributions would help justify the paper's novelty.

**Limitations:**

No, The paper has not sufficiently discussed limitations or societal impacts. The only mention is a single generic sentence in the Impact Statement (“none which we feel must be specifically highlighted here”).

**Strengths And Weaknesses:**

**Strengths:**

Fairly isolates effects of four latent-action strategies for the first time.

Clear empirical discoveries: Image-based latents are good at long-horizon tasks and action-based on motor complexity


**Weaknesses:**

The problem outlined by the paper is the lack of systematic comparison among fragmented latent-action integration strategies. While there are several good findings in the paper such as the clear formulation-task correspondence, the superiority of discrete token supervision over continuous embeddings, and reduced negative transfer in multi-task training, I am not sure if the novelty is enough for ICML as it seems the paper only compares some existing integration strategies under a unified baseline rather than introducing fundamentally new methods.

While the inclusion of real-robot experiments on the JAKA arm is great, the evaluation is limited to a small set of tasks. More real-world scenarios could strengthen claims.

The paper provides training hyperparameters and architectural specifics, which is positive, but there is no mention of code release or public repositories, potentially hindering replication efforts.

---

> ### Author Rebuttal · Authors · 2026-03-29
>
> We thank the reviewer for the positive recognition of our findings and address each point in turn.
>
> **Weakness 1 & Q2**: A clearer distinction between re-implementing existing ideas and original technical contributions is requested.
>
> The following components are original contributions:
>
> * LA-Align: a new method that applies implicit representation alignment to latent action supervision in VLA models, inspired by aligning 3D spatial signals with image representations (Li et al., 2025).
> * Action-based latent action model: a novel combination of FFT-based frequency encoding, temporal convolutions, and a latent consistency loss $\mathcal{L}_\text{mask}$. And both LA-Align and the action-based latent action model (LA-Tok) achieve results that outperform prior latent-action-based methods on LIBERO and leaderboard models on RoboTwin 2.0.
> * LA-Cond: instantiates conditioning pattern within a unified single-stage VLM backbone via causal attention masking, requiring no additional modules or stage separation.
>
> Additionally, even for strategies that build on existing paradigms, our implementation yields meaningful improvements: LA-Direct follows the same paradigm as UniVLA, yet our 2B-parameter implementation achieves 97.1% on LIBERO, surpassing UniVLA's 95.2% obtained with a 7B model.
>
> Prior works that relate to our strategies:
>
> * LA-Direct: is inspired by UniVLA, which supervises the VLM to predict latent action tokens in next-token prediction formulation.
> * Latent actions as intermediate reasoning for action prediction: prior works adopt this idea but couple it with specific training stages or additional architectural designs (LAPA, Moto-GPT, villa-x).
> * Action-based latent actions for action space alignment (Align-then-Steer).
>
> We will revise the paper to more explicitly delineate these contributions, and thank the reviewer for the suggestion.
>
> **Weakness 2**: The real-world evaluation is limited to a small set of tasks, and more real-world scenarios are requested to strengthen the claims.
>
> JAKA experiments are designed to probe capability boundaries : bowl stacking as a long-horizon task requiring sequential reasoning, and stain wiping as a contact-rich task involving a deformable object.
>
> We additionally designed three pick-and-place tasks, with OOD setting to require scene-level reasoning: *pick up the mango/sponge on the table into a bowl, and pick up the bottle of coke on the table*. For each task, 50 demonstrations were collected. The tabletop scene contains distractor objects (e.g., a Christmas hat, blocks, other bottles) in addition to the target object.
>
> We evaluate under two settings: ID (in-domain), where object positions and scene configurations are similar to training data; and OOD (out-of-domain), where we vary the target object position, randomly rearrange the scene layout, and introduce new distractor objects not seen during training.
>
> Success is measured as binary task completion, with 10 rollouts per model per task.
>
> | Method | Mango (ID) | Mango (OOD) | Sponge (ID) | Sponge (OOD) | Bottle (ID) | Bottle (OOD) | Avg. (ID) | Avg. (OOD) | Avg. (Total)|
> |:-------|:----------:|:-----------:|:-----------:|:------------:|:-----------:|:------------:|:---------:|:----------:|:----:|
> | Baseline | 1.0 | 0.2 | 0.6 | 0.2 | 0.7 | 0.0 | 0.77 | 0.13 | 0.45 |
> | LA-Align | 0.9 | 0.8 | 0.9 | 0.6 | 0.6 | 0.3 | 0.8 | 0.57 | 0.68 |
> | LA-Direct | 1.0 | 0.7 | 0.9 | 0.8 | 0.6 | 0.5 | **0.83** | 0.67 | **0.75** |
> | LA-Cond | 0.9 | 0.9 | 0.9 | 0.8 | 0.5 | 0.4 | 0.77 | **0.70** | 0.73 |
> | LA-Tok | 1.0 | 0.6 | 0.9 | 0.7 | 0.6 | 0.2 | **0.83** | 0.5  | 0.67 |
>
> These results are consistent with the main conclusions. LA-Tok improves ID success rates, particularly on tasks involving irregular (mango) and deformable (sponge) objects, consistent with its advantage on complex tasks. Under the OOD setting, where policies must reason about scene, image-based latent action strategies (LA-Align, LA-Direct, LA-Cond) demonstrate a clear advantage, consistent with our finding that image-based latent actions benefit tasks requiring scene-level reasoning, with directly predicting discrete latent tokens (LA-Direct) achieving the best overall performance.
>
> These additional experiments will be included in the revised paper. We thank the reviewer for the suggestion.
>
> **Weakness 3 & Q1**: The absence of code and data release plans is noted.
>
> We have submitted an anonymized code attachment alongside this submission, and for your convenience, it is also available at https://anonymous.4open.science/r/From_Pixels_to_Tokens-0009. And we will publicly release the full dataset.
>
> **Limitatioins**
>
> We kindly refer the reviewer to our response to Reviewer **t3Lm**, where we provide a detailed discussion of the limitations of this work.
>
> [1] Li, Fuhao, et al. "Spatial forcing: Implicit spatial representation alignment for vision-language-action model." arXiv preprint arXiv:2510.12276 (2025).

---

> > ### Author Rebuttal · Reviewer_ngJP · 2026-04-01
> >
> > It is still unclear whether the authors plan to release the code publicly. They have confirmed that the dataset will be released, but there is no mention of the code.

---

> > > ### Author Response · Authors · 2026-04-02
> > >
> > > Thank you for the follow-up. We confirm that the code, dataset, and model checkpoints will be open-sourced. The codebase has been prepared and converted to an anonymized repository (https://anonymous.4open.science/r/From_Pixels_to_Tokens-0009) for double-blind review compliance (the dataset is not included due to the 2GB storage limit of the anonymized platform). Once the review period ends, we will restore the full public GitHub repository, which will include the code, model checkpoints, and the real-world JAKA demonstration dataset (including the additional pick-and-place experiments reported in this rebuttal). We will also revise the corresponding sections in the paper.

---

### Official Review · Reviewer_NL3U · 2026-03-12

**Soundness:** 4
**Presentation:** 2
**Significance:** 3
**Originality:** 3
**Overall Recommendation:** 5
**Confidence:** 4

**Summary:**

The paper compares different latent action supervision methods on Vision-Language-Action (VLA) models, including a direct supervision method with action-based latent actions rather than Image based latent actions. Differing prior methods conflict in views of how latent actions effect VLA models with differing methods and approaches, resulting in a lack of understanding of how supervision choices effect VLA models from a shared baseline model. Evaluating on LIBERO and RoboTwin2.0 benchmarks along with Real-world evaluations showed while all latent supervised methods outperformed the base, Explicit Direct Decoding methods achieved the best performance across all. Within these, using Image-based latent actions yielded better performance on long action planning tasks (LIBERO-Long), whereas action based latent actions are better suited to complex motor tasks (RoboTwin2.0), uncovering a trade-off. Finally discrete supervision consistently outperforms continuous regression that still improves on baseline performance clearly demonstrating the advantage of latent action supervision.

**Compliance With Llm Reviewing Policy:**

Affirmed.

**Final Justification:**

Thanks to the authors for adequately addressing my questions, I am raising my score accordingly. It was also helpful to see author's responses to other reviews.

**Key Questions For Authors:**

1.	Figure 3 (a) Baseline shows VLM hidden layer embeddings corresponding to the image input embeddings being passed to the action head along with those corresponding to the action placeholders. However as 3.3 describes the action head as only receiving placeholder representations and proprioceptive state. Can the authors clarify whether there is a mistake in Figure 3 (a) or whether section 3.3 is wrong?
2.	In Section 5.4, isn’t the distinction between discrete tokens and continuous representations already well‑established? If not, particularly in the context of latent supervision, could you clarify why? For example, an MSE loss on continuous latent representations penalizes the model for deviating from a single ground‑truth target even when multiple valid latent actions exist, whereas a discrete formulation with a probability distribution can accommodate this variability by allowing multiple plausible latent outcomes.
3.	Appendix A clarifies “same training datasets used for policy learning”, what datasets are these?

**Limitations:**

Yes, the authors have adequately discussed the limitations.

**Strengths And Weaknesses:**

Soundness:
Claims are well supported by experimental results. Strengths and weakness of the methods used are adequately discussed. Technical explanation is correct and methods are appropriate.

Presentation:
Figures are generally well made and easy to follow as are tables and given equations. Writing is clear, eligible and easy to follow.
A short section or clarification of training data used is needed. Appendix D does clarify VLA training from all benchmark training data across all tasks. Appendix A clarifies “same training datasets used for policy learning” but what datasets this is referring to is not clear.
Figure 3 could benefit from including the action head for readability.
Figure 3 (a) Baseline shows VLM hidden layer embeddings corresponding to the image input embeddings being passed to the action head along with those corresponding to the action placeholders. However as 3.3 describes the action head as only receiving placeholder representations and proprioceptive state. If it were to receive all three; the baseline's worse performance compared to the new methods could be due to the lack of these embeddings and not the latent supervision.
Not sure what action head is used with LA-Align, thus assuming the same as base; may need slight clarification.
The paper's place within literature is clearly outlined explaining that their model focuses on the comparison of only the supervision policy methods themselves, providing an understanding and direct comparison of those methods.

Significance:
This paper addresses a lack of a definitive comparison of prior latent supervision methods to a baseline, providing insight into their differences. This helps to advance understanding and provides to further research two comparative best methods to build on (LA-Direct and LA-Tok).

Originality:
Demonstrates improved understanding with no novel new methods.
Prior works have not conducted a pure abolition of latent action supervision with the same latent action embeddings and head architecture. An understanding of discrete and continuous distribution in VLAs is further provided, which to my understanding is also not present in current literature.

---

> ### Author Rebuttal · Authors · 2026-03-28
>
> We thank the reviewer for the careful reading and the positive recognition of the paper's significance in providing a systematic comparison of latent supervision methods. We are glad that the identified best-performing strategies (LA-Direct and LA-Tok) are seen as useful foundations for further research. We address each concern below.
>
> **Q1**: Inconsistency between Figure 3(a) and Section 3.3.
>
> Thank you for catching this. The **figure is correct**; the error lies in Section 3.3 and the corresponding $f_{head}$ formulations throughout the paper. The action head for Baseline and all four LA variants receives three inputs: (i) the aggregated placeholder representations $h_t^\text{act}$ (or $h_t^\text{latent}$ under latent supervision), (ii) the robot proprioceptive state $s_t$, and (iii) the aggregated image token representations from the VLM backbone. This is consistent with our implementation, which has been submitted as an anonymized code attachment (for your convenience, the implementation is also available at https://anonymous.4open.science/r/From_Pixels_to_Tokens-0009). We will correct Section 3.3 and all $f_{head}$ formulations accordingly.
>
> LA-Align uses the same action head as other three varients and Baseline (available at [LA-Align implementation](https://anonymous.4open.science/r/From_Pixels_to_Tokens-0009/latentvla/models/vla/LA_Align.py), all five models use the same `L1RegressionActionHead` module).
>
> **Q2**: Is the discrete vs. continuous distinction already well-established?
>
> We agree this distinction has precedent in the tokenization literature. However, Section 5.4 addresses a more specific question: in the context of directly latent action supervision, should the VLM learn to map its representations into an external latent action space (cross-space regression), or internalize latent action semantics within its own token space (in-space token prediction)?
>
> Our results confirm that in-space token prediction consistently outperforms cross-space regression (+2.7% and +2.0% for LA-Direct and LA-Tok respectively), supporting a broader design principle: to equip a VLM with VLA capabilities, latent action semantics should be learned within the VLM's own representation space rather than bridged via external mappings. We attribute this to the fact that cross-space regression relies on a projection head to bridge two geometrically heterogeneous spaces, providing only indirect gradient signals to the VLM, whereas in-space token prediction forces the VLM to directly encode latent action semantics within its own representation space, consistent with its pre-training inductive bias. As the reviewer notes, the discrete formulation also naturally accommodates the multimodality of manipulation via a probability distribution over the codebook.
>
> **Q3**: What datasets are used for training the latent action model?
>
> The latent action models are trained(post-trained) offline on benchmark training data, used to annotate training trajectories, and then frozen during VLA policy training. Concretely:
>
> * LIBERO: public training splits from all four suites (Spatial, Object, Goal, Long).
>
> * RoboTwin 2.0 (main results): clean 50-trajectory per task for the evaluated 4-task subset.
>
> * RoboTwin 2.0 (mixed-training): clean 50-trajectory per task from evaluated 10 tasks.
>
> * JAKA real-world: 50 demonstrations per task.
>
> We will revise the corresponding sections in the paper to make this clearer and reduce potential misunderstanding.

---

> > ### Author Rebuttal · Reviewer_NL3U · 2026-04-01
> >
> > Thanks to the authors for adequately addressing my concerns.

---

### Official Review · Reviewer_t3Lm · 2026-03-18

**Soundness:** 3
**Presentation:** 3
**Significance:** 3
**Originality:** 3
**Overall Recommendation:** 5
**Confidence:** 3

**Summary:**

This work systematically investigates the latent action supervision in vision-language-action models. The authors perform experiments from two perspectives: (1) treating latent action as special images, and (2) unifying target space with latent actions. By comparing the performance of different strategies, the authors find that image-based representation benefit long-horizon reasoning, while action-based latent actions improve motor coordination.

**Compliance With Llm Reviewing Policy:**

Affirmed.

**Final Justification:**

The rebuttal resolved my concerns. It's a good paper. Therefore, I support acceptance.

**Key Questions For Authors:**

Please see weaknesses.

**Limitations:**

please provide limitations.

**Strengths And Weaknesses:**

# Strengths
1. The paper addresses an important problem in VLA models. Therefore, the work is well-motivated.
2. The experiments are comprehensive and well-designed. The authors compare different strategies for latent action supervision and analyze their effects on performance.
3. The paper is well-written and easy to follow.

# Weaknesses
1. This work is more of an empirical study rather than a novel method. Although the findings are interesting, it is more like an engineering effort to compare different strategies rather than proposing a new approach. Although I value the importance of this kind of work, I'm not sure whether it is suitable for a conference submission.

---

> ### Author Rebuttal · Authors · 2026-03-28
>
> We thank the reviewer for the positive recognition of our findings and address the novelty concern as follows.
>
> **Weakness**: The work is seen as an engineering effort to compare existing strategies rather than introducing novel methods.
>
> The core of this paper is indeed a systematic study, but one that yields three findings with direct practical value: (i) a formulation-task correspondence between latent action type and task type; (ii) directly supervising the VLM to predict discrete latent action tokens is the most effective integration architecture; and (iii) latent action supervision serves as an effective intermediate representation that substantially reduces negative transfer in multi-task joint training.
>
> Beyond the systematic study, viewing latent actions as intermediate representations between the VLM and action spaces naturally motivates new integration strategies. We propose LA-Align, inspired by 3D spatial signal fusion with VLMs (Li et al., 2025), and a novel action-based tokenizer combining FFT-based frequency encoding, temporal convolutions, and a latent consistency loss $L_{mask}$​. The corresponding variants LA-Align and LA-Tok achieve competitive results, outperforming prior latent-action-based methods on both LIBERO and RoboTwin 2.0.
>
> Finally, even for strategies that build on existing paradigms, our abstracted architectural instantiations consistently outperform their original implementations. For example, LA-Direct, which follows the discrete prediction paradigm of UniVLA, achieves 97.1% on LIBERO with a 2B model, surpassing UniVLA's 95.2% obtained with a 7B model. On RoboTwin 2.0, LA-Tok (78.0%) surpasses all entries on the public leaderboard including DP3 (71.0%) and $\pi_0$​ (64.0%).
>
> **Limitations**: please provide limitations.
>
> Our work has several limitations. First, our integration strategies are abstracted from mapping directions and prior works, and may not cover all possible supervision formulations. Second, while we adopt competent latent action models, it remains open whether further improving representation quality would amplify or narrow the observed performance gaps between strategies. Finally, due to limited experimental scale and platform availability, real-world evaluation is conducted solely on a single-arm robot (JAKA minicobo), and broader validation across more diverse robotic platforms would be desirable.
>
> [1] Li, Fuhao, et al. "Spatial forcing: Implicit spatial representation alignment for vision-language-action model." arXiv preprint arXiv:2510.12276 (2025).

---

> > ### Author Rebuttal · Reviewer_t3Lm · 2026-04-03
> >
> > Thank the authors for the rebuttal. I'll raise my score. Good luck.

---

### Decision · Program_Chairs · 2026-04-30

**Decision:**

Accept (spotlight)

**Comment:**

This paper presents work on vision-language-action models.  A systematic evaluation of latent actions in these models is conducted.  The reviewers appreciated the importance of the work (understanding the functioning of this important class of models), the thorough and convincing empirical data that back the evaluation, and the clarity of presentation.

The reviewers are unanimously in favour of accepting the paper.  The timely, important examination of VLA models and their internal workings are likely to generate new insights to bring the field forward.  The paper is well-written, solidly presented and motivated, and contains convincing empirical evidence to back the claims.  For these reasons the paper is recommended for acceptance.